# Learning Causal Models under Independent Changes

**Sarah Mameche**
CISPA Helmholtz Center
for Information Security
sarah.mameche@cispa.de

**David Kaltenpoth**
CISPA Helmholtz Center
for Information Security
david.kaltenpoth@cispa.de

**Jilles Vreeken**
CISPA Helmholtz Center
for Information Security
jv@cispa.de

## Abstract

In many scientific applications, we observe a system in different conditions in which its components may change, rather than in isolation. In our work, we are interested in explaining the generating process of such a multi-context system using a finite mixture of causal mechanisms. Recent work shows that this causal model is identifiable from data, but is limited to settings where the *sparse* mechanism shift hypothesis [1] holds and only a subset of the causal conditionals change. As this assumption is not easily verifiable in practice, we study the more general principle that mechanism shifts are *independent*, which we formalize using the algorithmic notion of independence. We introduce an approach for causal discovery beyond partially directed graphs using Gaussian process models and give conditions under which we provably identify the correct causal model. In our experiments, we show that our method performs well in a range of synthetic settings, on realistic gene expression simulations, as well as on real-world cell signaling data.

## 1 Introduction

Most of statistical learning theory rests on the assumption that all data is i.i.d. and comes from a single distribution. In reality, this assumption rarely holds: observations often stem from heterogeneous environments, different time domains, or interventional experiments. This results in non-i.i.d. data which may contain spurious correlations, such that naive analysis could be subject to serious learning bias [2]. Recent research tackles this problem via causal discovery, where given different context distributions, we want to discover robust, causal relationships that transfer to novel conditions [3]. Existing approaches however commonly discover only an equivalence class of the true model [4–9]. Here, we investigate under which conditions we can identify causal directions.

To determine causal orientations, Schölkopf et al. [1] proposed the Sparse Mechanism Shift (SMS) hypothesis, which asserts that distribution shifts are a consequence of a small number of causal mechanism changes. It relies on the idea that causal functions exhibit robustness over different contexts, whereas spurious correlations typically fail to generalize to modified conditions [3, 10, 11]. While the SMS assumption permits causal identification via counting of distribution shifts [12, 13], this approach is limited to settings where only a subset of the causal conditionals change, does not apply in the i.i.d. case, and needs *conditional independence* tests in practice, which suffer from a multiple testing problem and, depending on the use case, limited power [14, 15]. To address these issues, we propose a functional modeling approach based on *algorithmic independence* and give identifiability guarantees under general assumptions. Instead of counting mechanism changes, our approach identifies the model that achieves the most succinct lossless description of the data.

To illustrate this idea, we consider three variables in five environments (Fig. 1). The environments correspond to different conditions that we observe our system in, such as interventional experiments in genomics [16, 17]. In our example, there exists some mechanism responsible for generating $Y$ from its cause $X$ which we call $f$. It remains the same in Contexts 1-4, even when the underlying causal mechanisms and hence distributions of the covariates $X, Z$ differ ($f_1$), and only changes in

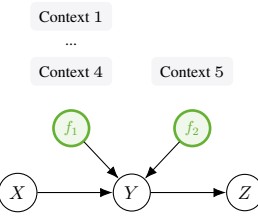
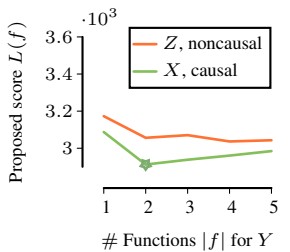
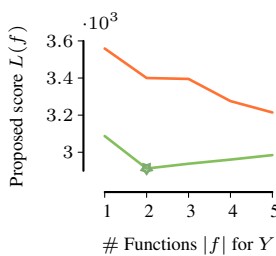

(a) Causal model for non-i.i.d. data.    (b) Covariate shift in one context.    (c) Covariate shift in all contexts.

Figure 1: Given non-i.i.d. data from different contexts, we are interested in the mechanism $f$ that generates a variable $Y$ and that may *change* in one or more contexts (a). We consider a varying number of causal functions $f$, modeled through a Gaussian process, in the causal (green) and anticausal (red) direction as potential models and measure how succinctly the models encode the data ($L$). Here, we find that $Y$ has cause $X$ and a small number $|f|$ of different mechanisms (green star). As the causal mechanism of $Y$ changes *independently* of the mechanisms generating $X$ and $Z$, this holds regardless of whether $X, Z$ undergo distribution shifts in one context (b) or in all contexts (c).

Context 5 as a result of a direct intervention on $Y$ ($f_2$). Intuitively, to most concisely describe the data, we will need two functions in the causal direction, fewer would underfit, and more overfit. To capture this, we propose a score that quantifies how succinctly a model can describe the observed distributions. We show in Fig. 1(b) that it identifies the ground truth (indicated by a star symbol).

To explain why an anti-causal model from $Z$ to $Y$ (red) fits the data less effectively than the causal one (green), we compare two cases. Fig. 1(b) depicts the case where we change the generating process of covariates $X$ and $Z$ in only one context, while in Fig. 1(c) we do so in all five contexts. Changing these processes does not directly affect how $Y$ is generated, and hence the true causal model permits equally succinct descriptions for both cases (green curve). In the anti-causal direction, however, such an invariant does not exist; in fact, by changing the distribution of $X$ and $Z$ in all five contexts we created a situation where five different functions are needed to describe $Y$ from $Z$, and none of these can as cleanly separate signal from noise as in the causal direction. As a result, we see much worse description lengths for the anti-causal direction (red curve). We conclude that the causal mechanism for $Y$ offers a succinct explanation because it is *independent* of the mechanisms for $X$ and $Z$. This connection between the independence of mechanisms and succinct descriptions will be the focus of our analysis.

Our contributions are as follows. We show how the algorithmic model of causation can precisely specify which kinds of causal models are permissible for a set of variables observed over multiple contexts. Based on this framework, we propose a scoring criterion under Gaussian process models [18]. We show that our score is theoretically sound by giving general conditions under which it identifies causal models beyond partially directed graphs. Based on our theory, we introduce the LINC approach for **L**earning causal models under **IN**dependent **C**hanges. In empirical evaluations, we confirm that LINC works in practice, and show that in contrast to competitors, it is robust in various settings including semi-synthetic gene expression data [19] as well as real-world protein-signaling data [20]. We include all proofs, code, and data in the supplementary material.

## 2    Causal Model and Assumptions

We begin by stating our assumptions and causal model. Throughout this work, we consider a set of continuous variables $X \in \mathbb{R}^d$. We observe these in multiple contexts $c \in \mathcal{C}$. For a given set of contexts $\mathcal{C}$, we can represent the assignment of samples to contexts using a categorical variable $C$. We write $X^{(c)}$ for a dataset in one context. Whereas the data points $x \in X^{(c)}$ are i.i.d. samples from a distribution $P(X \mid c)$, the overall data is not necessarily identically distributed. We aim to understand this data through a causal model, as we describe next.

## 2.1 Problem Setting and Problem Statement

We assume that we can describe the true causal structure over the observed variables by a directed acyclic graph (DAG) $G = (X, \mathcal{E})$ with node set $X$ and edges $(X_i, X_j) \in \mathcal{E}$ whenever $X_i$ causes $X_j$. As is standard in causal discovery [10], we assume that the relationship between $X_i$ and its parents is given by a function $f$,

$$X_i^{(c)} = f_i^{(c)}(\text{pa}_i^G, N_i) \text{ where } N_i \perp\!\!\!\perp (\text{pa}_i^G, C)$$

for each context $c$. We hereby assume causal sufficiency, i.e. that no unobserved variables exist. Whereas the structural equations depend on the context, the causal structure $G$ is shared in all contexts. Previous work shows that we can discover $G$ up to an equivalence class by integrating a context variable $C$ into conventional causal discovery methods. In particular, the $\psi$-Markov Equivalence class ($\psi$-MEC) of $G$, a variant of MECs to encompass interventions, is identifiable [6].

While graphical models are essential, we place emphasis on understanding the generating process of each variable. For a given variable $X_i$, we consider a mixture of $m$ different causal mechanisms $f$. Importantly, each may apply to one context or remain invariant over multiple contexts. To model this situation, let $\Pi_i = \{\pi_1, ..., \pi_m\}$ be a partition of $\mathcal{C}$ into disjoint, nonempty sets $\pi$ such that

$$f_i^{(c)} = f_i^{(c')} \text{ for } \Pi_i(c) = \Pi_i(c') ,$$

where $\Pi_i(c)$ is the unique set $\pi \in \Pi_i$ containing $c$. We thus write $f_i^{(\pi)}$ for the invariant function $f_i^{(c)}$ when $c \in \pi$, and similarly $X^{(\pi)}$ for the pooled data from all contexts $c \in \pi$. To ease notation, we write its distribution as $P(X \mid \pi) := P(X \mid C \in \pi)$. This results in a set of functions

$$f_i^{(\Pi_i)} = \{f_i^{(\pi_1)}, ...f_i^{(\pi_m)}\} .$$

We obtain such a partition of the contexts for each variable and denote the set over those partitions by $\mathbf{\Pi} = \{\Pi_i : X_i \in X\}$. Our structural equations can thus be written as

$$X_i^{(c)} = f_i^{(\pi)}(\text{pa}_i^G, N_i) \text{ for } c \in \pi \tag{1}$$

for each $\pi \in \Pi_i$, with independent noise $N_i \perp\!\!\!\perp (\text{pa}_i^G, C)$. We remark that by assuming the noise $N_i$ to be independent of the environment, we follow how related work characterizes environments [7, 9, 13] as differing in distribution due to changes in the conditionals $P(X_i \mid \text{pa}_i^G)$, independently of the exogenous noise variables $N_i$, for example, due to a soft intervention on $X_i$.

With the above, we can now informally state the problem that we want to solve.

**Problem Statement** (Informal). *Given data $X$ over environments $\mathcal{C}$ we want to find*

- *a (sparse) graph $G$*

- *a set $\mathbf{\Pi}$ of partitions $\Pi_i$ of $\mathcal{C}$ into (few) disjoint sets*

- *a set of (simple) functions $f_i^\pi$*

*describing the causal mechanism generating our data $X$.*

To formalize this problem, we next explain the principle of independence of causal mechanisms.

## 2.2 Algorithmic Independence of Causal Mechanisms

We base our approach on the principle of Independence of Causal Mechanisms (ICM, [10]). Different types of independence have been proposed in the literature. Most previous work considers statistical independence [4–9, 21], but the conditional independence constraints present in observational data only permit the identification of DAGs up to a MEC unless we make restrictive assumptions [3, 13, 22]. This has sparked interest in algorithmic independence as an alternative formulation [23–27]. We introduce this concept here and extend it to multi-context distributions.

With the idea that causal mechanisms offer simple explanations of complex phenomena, Janzing and Schölkopf [23] postulate that they correspond to simple programs computing each variable $X_i$ from its causes $\text{pa}_i^G$, where programs generating different variables $X_i, X_j$ do not share information.

To make this precise, we revisit several concepts. Kolmogorov complexity $K(x)$ defines the length of the shortest program $p$ that computes a string $x \in \{0, 1\}^*$ on a universal Turing machine $\mathcal{U}$ and halts [28]. For a distribution $P$, Kolmogorov complexity is the length of the shortest program able to uniformly approximate $P$ arbitrarily well,

$$K(P) = \min_{p \in \{0,1\}^*} \left\{ |p| : \forall y \, |\mathcal{U}(p, y, q) - P(y)| \leq \frac{1}{q} \right\} .$$

Using $K$ and towards defining algorithmic independence, we define algorithmic mutual information [23]. For two binary strings $x, y$, the algorithmic mutual information between $x, y$ is given by $I_A(x; y) = K(x) + K(y) - K(x, y)$. For distributions $P, Q$, we can state it as follows.

**Definition 2.1** (Algorithmic Mutual Information [28]). The algorithmic mutual information between two distributions $P$ and $Q$ is given by

$$I_A(P; Q) := K(P) - K(Q \mid P^*)$$

where $P^*$ is the shortest algorithmic description of $P$.

Two strings $x, y$ are algorithmically independent if $I_A(x; y) \overset{\pm}{=} 0$ holds (up to an additive constant that is independent of $x, y$), where we write $I_A(x_1; ...; x_n)$ if the corresponding strings are pairwise independent. Analogously, a set of distributions $P_i$ are jointly independent when

$$I_A(P_1; \ldots; P_n) \overset{\pm}{=} 0 .$$

When the mutual information $I_A$ between different causal conditionals vanishes, we can characterize them as independent. The algorithmic causal Markov Condition [23] states precisely this.

**Postulate 2.2** (Algorithmic Markov Condition for Changing Mechanisms). *Let $(G, \mathbf{\Pi})$ be a causal model with DAG $G$, partitions $\mathbf{\Pi}$, and structural equations as in Eq. 1. Then $(G, \mathbf{\Pi})$ is a valid causal hypothesis only if the following equivalent conditions hold,*

1. *All causal mechanisms $f$ in the model are jointly algorithmically independent. That is,*

$$I_A(\{ P(X_i \mid pa_i^G, \pi) : i \in \{1, ..., n\}, \pi \in \Pi_i \}) \overset{\pm}{=} 0 . \tag{2}$$

2. *The causal mechanisms $f$ in the model describe the observed distribution most concisely. That is, the shortest description of $P(X)$ across all contexts $c \in \mathcal{C}$ is given by writing each $X_i$ as a function of $pa_i^G$ in each group $\pi \in \Pi_i$,*

$$K(P(X)) \overset{\pm}{=} \sum_{X_i} \sum_{\pi \in \Pi_i} K(P(X_i \mid pa_i^G, \pi)) \tag{3}$$

*where $\overset{\pm}{=}$ denotes equality up to an additive constant.*

The inner term in Eq. (2) ranges over the conditionals $P(X_i \mid \text{pa}_i^G, \pi)$ for each variable $X_i$ in each group of contexts $\pi \in \Pi_i$, where we omit indices of $\pi$ for ease of notation. Overall, Postulate 2.2 states that the causal mechanisms correspond to programs that do not share any information (Eq. (2)) and that describe the observed distributions most concisely (Eq. (3)).

We will use the formulation in Eq. (3) which directly uses Kolmogorov complexity of conditionals. Due to the halting problem, we cannot compute Kolmogorov complexity for arbitrary programs [28], but by restricting the class to be tractable, we can approximate it from above. A statistically well-founded manner of doing so is Minimum Description Length (MDL) [29]. For a fixed class $\mathcal{H}$ of models, MDL defines a description length $L$ of encoding a dataset $X$ together with its optimal model,

$$L(X \mid \mathcal{H}) = \min_{h \in \mathcal{H}} \left( -\log P(X \mid h) + L(h) \right) .$$

Regarding the choice of $\mathcal{H}$, we are interested in functions $f$ that are both expressive but also allow us to measure their description length. Gaussian processes naturally fulfill these requirements [18, 29], making them a fitting choice for our approach. We briefly introduce them next.

## 2.3 Gaussian Process Functional Model

A Gaussian process (GP) is a collection of random variables $\mathcal{X}$, any finite number of which have a joint multivariate Gaussian distribution [18]. Each variable represents the value of a function $f$ at a single location $x$. We write functions drawn from a GP as

$$f \sim \mathcal{GP}(m(x), \kappa(x, x'))$$

for a mean function $m(x) = \mathbb{E}[f(x)]$ and covariance kernel $\kappa(x, x') = \mathbb{E}[(f(x) - m(x))(f(x') - m(x'))]$. Viewed differently, a GP can also be expressed as a linear regression $f = \sum \alpha_i \phi_i$ with potentially infinitely many basis functions $\phi_i$. Given finitely many data points $(x_i, y_i)$, we can write predictions as a linear combination of a finite number of kernel functions $\kappa(x_i, \cdot)$ centered on the observed points $x_i$ [18],

$$f(x) = \sum_{i=1}^{n} \alpha_i \kappa(x_i, x) , \tag{4}$$

with $\alpha = (K + \sigma^2 I)^{-1} y$, where $K$ is the Gram matrix of $\kappa$ with $K_{ij} = \kappa(x_i, x_j)$, and $\sigma^2$ a parameter. This formulation will be useful when we consider the description length of GPs later.

Regarding the richness of the function class, given a kernel $\kappa$, one can construct a reproducing Hilbert space $\mathcal{H}_\kappa$, which is the smallest Hilbert space containing all finite expansions of the form in Eq. (4). For some kernels, such as the radial basis function (RBF) kernel $\kappa(x_i, x_j) = \lambda \exp(-\frac{1}{2l} \|x_i - x_j\|_2^2)$, $\lambda > 0$ which we consider here, it is known that $\mathcal{H}_\kappa$ is dense in the space of continuous functions [18]. That is, we can approximate these functions up to arbitrary precision using a GP [30]. Using GPs, we now state our functional model as follows.

**Assumption 1** (Functional Model with Changing Mechanisms). Given a DAG $G$ and partitions $\Pi$, we assume the functional model

$$X_i^{(c)} = f_i^{(\pi)}(\mathrm{pa}_i^G) + N_i \text{ for } c \in \pi, \tag{5}$$

where $\pi \in \Pi_i$ is a group, $f_i^{(\pi)} \in \mathcal{H}_\kappa$ is a GP, and $N_i$ is independent noise $N_i \perp\!\!\!\perp (\mathrm{pa}_i^G, C)$.

We point out that the above makes an additive noise assumption as is standard in GP modeling. Having stated the ICM assumption and our functional model, we are ready to introduce our approach.

# 3 Learning Independent Changes with LINC

We now introduce a score to instantiate Eq. (3) for GPs and give an analysis of its properties.

## 3.1 Measuring the Complexity of GP Models

To define the ability of our model to succinctly describe a given dataset, we need to address the complexity of a GP $f$. Despite the richness of $\mathcal{H}_\kappa$, the complexity of any $f \in \mathcal{H}_\kappa$ can be readily measured by its squared norm $\|f\|_\kappa^2$. For functions inferred from finite amounts of data, as in Eq. (4), we can compute this norm $\|f\|_\kappa^2$ from the quantities $K$ and $\alpha$ as [29]

$$\|f\|_\kappa^2 = \alpha^\top K \alpha .$$

Consider now the model class $\mathcal{H}_\kappa$ of GPs and a target variable $X_i$. For a given subset $X_S \subseteq X$ of predictors, we can fit a GP regression $f : X_S \mapsto X_i$. Kakade et al. [31] show that the Bayesian MDL score for this GP model is given by

$$L(X_S, X_i \mid \mathcal{H}_\kappa) = \min_{f \in \mathcal{H}_\kappa} \left( -\log P(X_i \mid X_S) + \|f\|_\kappa^2 \right) + R(X_S) . \tag{6}$$

The score balances the data fit of the GP regression model $f$ with its complexity, $\|f\|_\kappa^2 = \alpha^\top K \alpha$. The remaining penalty $R(X_S)$ is independent of the minimization over $f$,

$$R(X_S) = \tfrac{1}{2} \log \det(\sigma^{-2} K_S + I) ,$$

where $K_S$ is the Gram matrix when $\kappa$ is applied to $X_S$. We write $L(f) = L(X_S, X_i \mid \mathcal{H}_\kappa)$ when the context is clear, and $L(G, \Pi)$ as the summed description length for a given causal model. By Eq. (3), we choose the model with the smallest description length as our causal hypothesis.

**Problem Statement** (Formal). *Given data $X$ over environments $\mathcal{C}$ we want to find the graph $\widehat{G}$, partitions $\widehat{\mathbf{\Pi}}$ and mechanisms $f_i^{(\pi)}$ minimizing the score L,*

$$\widehat{G}, \widehat{\mathbf{\Pi}} = \arg\min_{G, \mathbf{\Pi}} L(G, \mathbf{\Pi}) \ . \tag{7}$$

To show that we recover the correct causal structure with this objective, we give identifiability results.

## 3.2 Identifiability Results

Here, we give conditions under which we can identify the true causal model using our score function. We begin by showing that we can discover the causal DAG in the correct Markov Equivalence class (MEC) under the following general assumptions.

**Assumption 2** (Sufficient Capacity). *For each $i$ there exist functions $f_i^{(\pi)} \in \mathcal{H}_\kappa$ s.t. Eq. (5) holds.*

This recalls the assumption that our functional model is well-specified.

**Assumption 3** (Causal Minimality). *For all $\pi \in \Pi_i$, $f_i^{(\pi)}$ is not constant in any of its inputs.*

The above is a toned-down version of the faithfulness condition, ensuring that every parent of $X_i$ has an influence on $X_i$. Without this, it would be impossible to identify the complete set of parents.

**Assumption 4** ($\Pi$-Faithfulness). *If $\pi' \neq \pi \in \Pi_i$ then $f_i^{(\pi)} \neq f_i^{(\pi')}$.*

This assumption guarantees the uniqueness of each partition. Without it, interventions would not necessarily lead to changes in causal mechanisms, and make identification thereof impossible.

With this, we can state our first identifiability result as follows.

**Theorem 3.1** (Identification of MECs). *Let Assumptions 1-4 hold and let $\lambda$ be sufficiently small. Then the estimates $\widehat{G}, \widehat{\mathbf{\Pi}}$ that minimize our score L,*

$$\widehat{G}, \widehat{\mathbf{\Pi}} = \arg\min_{G, \mathbf{\Pi}} L(G, \mathbf{\Pi}) \ ,$$

*recover a DAG $\widehat{G} \sim G^*$ in the Markov Equivalence class of the true DAG $G^*$, as well as the true partitions $\Pi_i^*$ for each node $i$ with probability 1*

$$P(\widehat{G} \sim G^*, \widehat{\mathbf{\Pi}} = \mathbf{\Pi}^*) = 1$$

Next, we show that under independent mechanism shifts, we can also identify the fully directed $G^*$. To correctly identify the direction of all edges in the discovered network, we need the following assumption that the causal mechanisms are independent of each other and change independently.

**Assumption 5** (Independent Mechanism Shift). *For all $i \neq j : \Pi_i(C) \perp\!\!\!\perp \Pi_j(C)$.*

We can now show that as we observe more contexts, $|\mathcal{C}| \to \infty$, our score identifies the correct DAG.

**Theorem 3.2** (Identifiability from independent changes). *Let Assumptions 1-5 hold. Then for $\lambda$ sufficiently small, where $\lambda$ is a hyperparameter bounding the score differences of $\widehat{G}, G^*$, we can identify the causes $pa_i^G$ and partition $\Pi_i$ of each $X_i$ in the causal model $(G^*, \mathbf{\Pi}^*)$ with probability 1,*

$$\lim_{|\mathcal{C}| \to \infty} P(\widehat{G} = G^*, \widehat{\mathbf{\Pi}} = \mathbf{\Pi}^*) = 1$$

*In particular, the fully directed network $G^*$ is identifiable.*

The requirement $|\mathcal{C}| \to \infty$ is necessary to turn the assumed independence of causal conditionals across different contexts into an empirically testable statistical criterion. That is, to guarantee that the model is indeed identifiable, we need either the distribution assigning contexts to partitions, or a guaranteed representative sample from the distribution, which we only obtain in the limit. Essentially, this is a frequentist approach, allowing us not to require the distributions for the partitions themselves.

We can, however, provide further probabilistic guarantees for finite numbers of contexts. To be able to reason about probabilities of different causal structures, we need to ensure that as we observe more contexts $|\mathcal{C}|$, the number of sets in the partition remains fixed. We hence assume that for each $X_i$, the contexts fall inside one of a fixed number sets $B_{i,1}, ..., B_{i,k_i}$ which can be thought of as bins that fix the sizes of our partitions, and thus bound the number of mechanism shifts per variable.

**Assumption 6** (Fixed Partition Sizes). For each $X_i$, there exists a finite number of sets $B_{i,1}, ..., B_{i,k_i}$ such that $p_{ij} := P(C \in B_{i,j}) > 0$.

The assumption can be understood in a frequentist sense, in that on average each possible causal mechanism for each variable occurs in at least a constant fraction of all samples. For example, if we study the differences between treatments in different hospitals, then as we obtain more data, these data would be obtained from the same set of hospitals, rather than by the addition of new hospitals. With this, we are able to give a tighter bound as follows.

**Theorem 3.3** (Identifiability in the finite context setting). *Let the Assumptions of Theorem 3.2 as well as Assumption 6 hold. Let $p = \min_{i,j} p_{ij}$ be the minimum probability for a context to fall into one of the bins $B_{i,j}$. Then if $|\mathcal{C}| \geq -2p^{-2} \log(p) > 0$ we have*

$$P(\widehat{G} = G^*, \widehat{\mathbf{\Pi}} = \mathbf{\Pi}^*) = \Omega\left(\exp\left(-|E| e^{-c}\right)\right),$$

*where $c = p^2 |\mathcal{C}| + 2 \log p \geq 0$ and $|E|$ is the number of edges in $G^*$.*

The dependence of the bound on $|E|$ is reasonable. The sparser the graph $G^*$, the fewer edges that need to be directed in its MEC. Note furthermore that the fixed size of the partitions is likely not strictly necessary, and as long as $p$ decays sufficiently slowly, Theorem 3.3 will still likely hold. We also note that the bounds above apply to the case where we have no active control over intervention targets, i.e. when mechanism shifts occur randomly. While active intervention design allows identification using a smaller number of interventions [32], we often cannot ensure full interventional control in practice; for example, gene editing technologies such as CRISPR-Cas gene-editing are known to have severe off-target effects [7, 16, 17] that may not be known a priori.

In what follows, we describe how to put our theoretical insights into practice.

### 3.3 Discovering Independent Changes with LINC

In summary, our approach, called LINC (**L**earning causal models under **IN**dependent **C**hanges), aims to discover a mixture of mechanisms that together most succinctly describe multi-context data.

To do so, we discover the causal DAG $G$ and partitions $\mathbf{\Pi}$ that minimize Eq. (7). To compute $L$ for a given model $(G, \mathbf{\Pi})$, we consider each causal mechanism $f^{(\pi)}$ in the model and fit a GP regression predicting $X_i$ from its causes $\text{pa}_i^G$. We compute the MDL description length of this GP as in Eq. (6). We do so in each group of contexts in $\Pi_i$ and each variable in $G$ to obtain the overall score $L(G, \mathbf{\Pi})$.

To discover the partitions for a known graphical model $G$, we suggest an exhaustive search whenever feasible, given that the number of different contexts that we have access to may be small in the typical use case [20]. As exhaustive search over partitions becomes prohibitive for larger numbers of contexts, we propose a greedy variant in Appendix B. It relies on the no-hypercompression inequality [29] that allows defining a scoring function for pairs of contexts. This pairwise measure can be used together with a clustering approach to discover a partition of the contexts in a greedy manner.

We also face the search for a DAG $G$. Given the large search space over causal DAGs, existing algorithms are of a greedy nature, such as the score-based greedy equivalence search (GES, 33) and its generalized version GGES that can be used with general scoring functions so long as they are locally consistent [15]. In principle, we can use the MDL score $L$ as a plug-in estimator in GGES to discover a MEC of the true graph. However, as there exist many greedy approaches to discover a MEC of $G$ from multiple contexts [4–9], whereas discovering relationships beyond MEC is relatively underexplored [12, 13], we put our emphasis on the latter and discover the best causal DAG within a given Markov equivalance class in our experiments, following the literature [13].

Finally, we also adress the computational burden associated with our score $L$ due to the sample complexity of GP regression. To alleviate this, we leverage the Random Fourier Feature (RFF) approximation, which allows us to express the GP kernel through a Fourier transform [34]. We adapt our scoring function accordingly to use the approximated RFF kernel. We call our main approach LINC$_{\text{GP}}$ and include a comparison to the variant LINC$_{\text{RFF}}$ in our evaluation.

We provide the details of our algorithms, a computational complexity analysis, and the adaptation to RFFs in the supplementary material. In the remainder of our work, we focus on evaluating LINC empirically against relevant related work, which we survey in the following section.

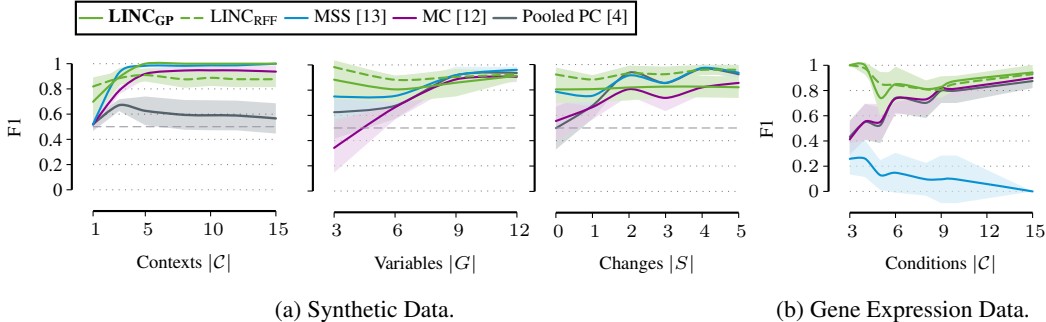

(a) Synthetic Data.    (b) Gene Expression Data.

Figure 2: *LINC discovers causal models beyond Markov Equivalence.* We report the F1 scores for discovering the causal direction of each edge in a random DAG $G$, where shading shows confidence regions and dashed horizontal lines illustrate random guessing. In (a), we consider synthetic data over different contexts $\mathcal{C}$ where $|S|$ variables per context undergo mechanism shifts, starting from $|\mathcal{C}| = 5, |G| = 6$ and $|S| = 2$. In (b), we mimic real-world gene expression dynamics by simulating data with SERGIO [19] from a random network $G$ over $|\mathcal{C}| - 1$ genes, which we observe in conditions $\mathcal{C}$ with a knockout intervention on one gene per condition.

## 4 Related Work

Discovering cause-effect relationships is of great interest in many applications, and there exists a growing literature on how we can do so from observational data [10]. Most well-known methods can be categorized either as constraint-based, such as the PC algorithm [35], or score-based, such as GES [33]. Given purely observational and identically distributed data, it is however impossible to make definitive statements about the causal structure [10]. Non-i.i.d. data carries information about how the system responds to external intervention or noise shifts, and hence often allows to infer causal directions more accurately [6, 36]. We summarize approaches to do so here.

A prominent idea in causal inference from multiple contexts is that causal mechanisms remain stable, i.e., *invariant*, under distribution shift [37] and there exist several causal discovery methods that use invariance as an asymmetry between causal and non-causal orientations [3, 38–40]. Invariance may be violated when variables are subject to interventions, as is common in biology [20]. Methods that can discover causal relationships from a combination of observational and interventional data [41–43] often assume known intervention targets, which is not always realistic [44]. Hence, recent research considers multiple contexts with uncertain interventions or *mechanism changes*. For example, Huang et al. [4] show how to apply the PC algorithm jointly to multiple contexts by introducing a context variable. There are similar approaches with different instantiations [5–9], all of which however discover causal MECs. A common approach to go beyond Markov Equivalence is *functional modelling* [15, 26, 25, 45, 46], but all existing methods that we are aware of are tailored to i.i.d. data.

Finally, recent works tackle causal discovery beyond Markov Equivalence from non-i.i.d. data under the *sparse mechanism shift* (SMS) [1] hypothesis. The Minimal Changes (MC) method [12] and the Mechanism Shift Score (MSS) [13] both count the number of distributional changes across contexts to infer edge directions within a MEC. MC assumes linear models, MSS is nonparametric. As our contribution, we show that regularized scoring functions, such as our MDL-based score, can be used toward the same goal. These scores have been shown to make use of the independence noise asymmetry on i.i.d. data [25, 47], which suggests that LINC may be more widely applicable than the SMS scores. As MC and MSS are most related to LINC, they will be the focus of our evaluation.

## 5 Evaluation

We consider synthetic, semi-synthetic, and real-world data to empirically evaluate LINC and how its performance relates to the state-of-the-art. We make the code and datasets available in the supplement.

**Experimental Setup**    In our experiments, we consider both synthetic data from a known functional model class, as well as more realistic data following the stochastic differential equations behind gene regulatory networks [19]. Throughout, we use an Erdös Rènyi DAG model to sample graphs $G$.

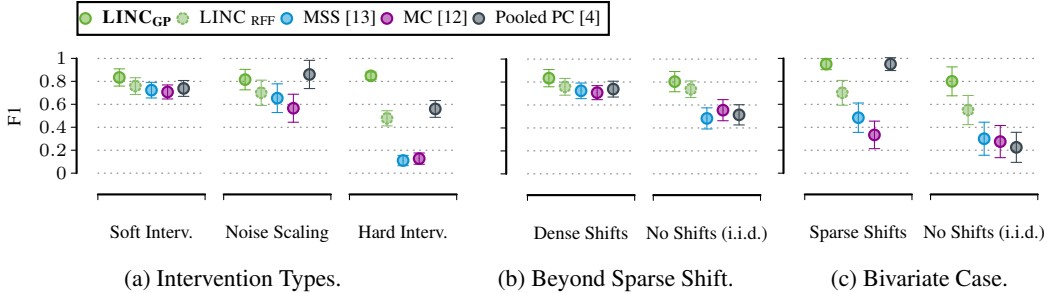

Figure 3: *LINC is robust in different settings.* We show which methods can discover edges beyond Markov Equivalence robustly. We consider (a) soft and hard interventions on the mechanisms in Eq. 8, (b) dense shifts compared to i.i.d. data, where all ($|S| = |\mathcal{C}|$), resp. none ($|S| = 0$) of the variables change, and (c) bivariate causal discovery in the base ($|S| = 2$) and i.i.d. case ($|S| = 0$).

For the synthetic data, following the literature [4, 13] we generate data in multiple contexts using

$$X_i^{(c)} = \sum_{j \in \mathrm{pa}_i^G} \omega_{ij}^{(c)} f_{ij}(X_i^{(c)}) + \sigma_j^{(c)} N_j^{(c)} \, , \tag{8}$$

with weights $\omega_{ij}^{(c)} \sim \mathcal{U}(0.5, 2.5)$, where noise is either uniform or Gaussian with equal probability. We sample the causal functions $f$ from $\{x^2, x^3, \tanh, \mathrm{sinc}\}$. In each context, we choose a random subset $S \subseteq \{0, ... |G|\}$ of variables that undergo a mechanism shift, which is either a soft intervention by re-sampling from Eq. (8), noise scaling through $\sigma \sim \mathcal{U}(2, 5)$, or a hard intervention via $w_{ij} = 0$.

In addition, we simulate data with SERGIO [19] to generate single-cell expression data according to a gene regulatory network while modeling the real-world dynamics of gene transcription and regulation using stochastic differential equations. Using a random DAG $G$ as the reference network, we generate one observational dataset as well as $|G|$ interventional datasets, in each of which we keep a different gene fixed to mimic a knockout intervention [48].

**Baselines**    As a baseline for causal discovery up to Markov Equivalence, we include Pooled PC [4] which pools data from all contexts, augments the system with a categorical variable $C$ that encodes the context and applies the PC algorithm to discover a causal DAG. As we are interested in evaluating whether LINC can orient edges beyond the MEC, we also compare against the linear Minimal Changes approach (MC) [12] and the nonlinear Mechanism Shift Score (MSS) [13] that are based on mechanism shift counting. We instantiate MSS with the Kernel Conditional Independence (KCI) test and use the soft score variant, which Perry et al. [13] show to consistently outperform competitors.

**Causal discovery beyond MECs**    As our main experiment, we evaluate whether LINC can discover causal edge orientations in non-i.i.d. data. We sample DAGs $G$ of size $|G| = 6$ with edge density $p = 0.3$ and generate data in $|\mathcal{C}| = 5$ contexts, with $|c| = 500$ samples and $|S| = 2$. We start from the correct MEC, use each method to discover a DAG, and report F1 scores over edge directions.

We report the results in Fig. 2(a). As pooled PC discovers a MEC jointly from all contexts, its F1 scores tend to 0.5, which can be achieved by not orienting any edges in the MEC as this results in recall 0.5 on expectation (dashed lines). In contrast, MSS, MC and LINC discover all edge orientations given sufficiently many contexts $|\mathcal{C}|$, variables $|G|$, and changes $|S|$. The differences are most pronounced for small DAG sizes, resp. contexts. In particular, only LINC learns additional edge directions from i.i.d., single-context data since the competitors rely purely on mechanism shifts. The RFF variant of LINC works surprisingly well but is ultimately similar to GPs for large $|\mathcal{C}|$.

Overall, LINC compares favorably against its competitors in all settings. We defer results with varying edge density $p$ and sample size $|c|$, as well as scalability with $|\mathcal{C}|$ and $|G|$ to Appendix B.

**Gene regulatory networks**    Next, we evaluate the methods on gene expression data that we generate with SERGIO. We show the F1 scores on edge directions in Fig. 2(b) depending on the number of conditions $|\mathcal{C}|$. While MSS is the closest competitor to LINC on synthetic data with *soft* interventions (Fig. 2(a)), the *hard* interventions in this experiment result in poor performance (Fig. 2(b)). Despite

assuming linearity, MC performs well, which suggests that it detects mechanism shifts even when its functional model is likely misspecified. It performs comparably to Pooled PC, where both benefit from an increased number of variables and conditions. LINC outperforms the competitors, the difference being most pronounced when it observes only few conditions.

**Causal discovery beyond SMS**  Besides confirming that LINC can handle different generating processes, we are also interested in its robustness w.r.t. different types of distribution shifts that may occur. In Fig. 3(a), we compare the soft interventions considered so far with noise scaling and hard intervention. To confirm our observations that LINC generalizes to settings where the sparse shift principle is not applicable, we compare dense shifts $|S| = |\mathcal{C}|$ to the case without any changes $|S| = 0$ in Fig. 3(b). As expected, the competitors roughly match the base MEC in the i.i.d. case, whereas LINC performs well. This observation is in line with recent theoretical results [25] that MDL-based scoring functions can exploit the additive noise asymmetry [45, 49] on i.i.d. data. Lastly, LINC also performs remarkably well compared to the baselines for causal direction determination between pairs of variables, as we show in Fig. 3(c).

**Cell signalling pathways**  Finally, we evaluate LINC on real-world data over eleven proteins and phospholipid components, with the goal of mapping their signaling pathways in human immune cells [20]. To properly understand how the components of such a system interact, it can be indicative how the system responds to stimulatory and inhibitory cues, which Sachs et al. [20] added to the system in multiple experiments. We run LINC and competitors on data from all contexts, without providing information about the interventions. We show the best DAG that LINC identifies within the true MEC in Fig. 4. We color edges that Sachs et al. [20] report as causal

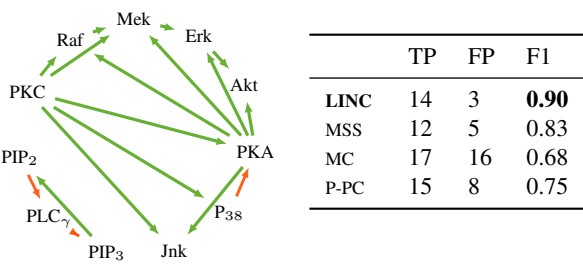

| | TP | FP | F1 |
|---|---|---|---|
| **LINC** | 14 | 3 | **0.90** |
| MSS | 12 | 5 | 0.83 |
| MC | 17 | 16 | 0.68 |
| P-PC | 15 | 8 | 0.75 |

(a) DAG with LINC.          (b) Edge Directions.

Figure 4: *Cell Signalling Pathways.* We show the DAG that LINC estimates for the cell cytometry data (a) in comparison to competitors (b). Green links are causal (TP) and orange edges are anticausal (FP) according to the literature [20].

in green and edges with opposite orientation in orange. Except for three, LINC discovers all directions correctly (Fig. 4(a)), leading to the highest performance compared to the competitors (Fig. 4(b)).

## 6 Discussion and Conclusion

In this work, we consider causal modeling of data from multiple environments where unknown causal mechanism changes may exist. Our interest in this setting is motivated by the prevalence of distribution shifts and heterogeneity in real-world applications, as well as by the need to analyze interventional data in biology, where modern gene editing and measurement technologies allow collecting large-scale experimental data [50]. Here, we showed that viewing independence of causal mechanisms from an information-theoretic perspective allows us to analyze such data in a unifying way. With LINC, we proposed a nonlinear functional modeling approach using Gaussian processes. We showed theoretically that we can use it to identify causal models with independent changes, and demonstrated in practice that it is applicable to settings beyond those where the sparse mechanism shift principle applies, and that it gives insight into biological data with unknown intervention effects.

While we focused on theoretical guarantees in this work, developing efficient algorithms for causal discovery is an important aspect of future work. Given the shift of research attention in causal discovery from i.i.d. to non-i.i.d. data, we face not only the problem of efficient search for DAG structures but in addition the problem that there may be multiple causal mechanisms per variable, which adds an additional layer of complexity. Towards efficient search for partitions, we propose search algorithms inspired by clustering in the supplementary, and establishing approximation guarantees for these methods is a worthwhile direction of future work. Other future directions include extending our theory to latent confounding, considering the situation where the split of the data into different contexts is not known to us, or applying our insights to causal representation learning.

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

# A Theory

## A.1 Background

We assume familiarity with Markov Equivalence of DAGs in the i.i.d. case, and refer to Lauritzen (1996) and Pearl (2009) for detailed definitions.

For non-i.i.d. or interventional distributions, identifiability of DAGs extends to an *interventional* Markov Equivalence class (iMEC) [36, 6]. This holds for general interventions targeting specific nodes in certain contexts. That is, an intervention targeting the nodes $X_i$ for $i \in I$ corresponds to the interventional distribution $P^{(I)}$, which factorizes as follows,

$$P^{(I)}(X) = \prod_{i \in I} P^{(I)}(X_i \mid \mathrm{pa}_i^G) \prod_{j \notin I} P^{\emptyset}(X_j \mid \mathrm{pa}_j^G)$$

where $P^{\emptyset}$ is the observational distribution [36]. Jaber et al. (2020) generalized the above to address the case where the observational context is unknown, and formally define the MEC that one can identify from multiple interventional distributions with unknown targets as the $\psi$-MEC [6].

In our work, we take a different perspective where instead of interventions with specific targets, we consider mechanisms that are invariant in a group of contexts or change independently. Similarly to the graphical representations of interventions through latent variables $C$ [7, 6], we can also represent our partitions as latent variables $\widehat{\Pi}_i(C)$. We will shortly show that we can discover both a DAG in the correct Markov Equivalence class as well as the correct partition for each node.

To do so, an important property of our MDL score that we will need is information consistency [29, 31] according to the following definition [52].

**Definition A.1** (Information Consistency). A prediction strategy $Q$ is information consistent with respect to a competitor space $\mathcal{F}_{\mathrm{comp}}$ if and only if

$$\lim_{n \to \infty} \tfrac{1}{n} \mathbb{E}[D_{\mathrm{KL}}\big(P(Y_{\leq n} \mid f, X_{\leq n}) \,\|\, Q(Y_{\leq n} \mid X_{\leq n})\big)] \to 0$$

holds for all $f \in \mathcal{F}_{\mathrm{comp}}$.

We refer the reader to Kakade et al. (2005) for a proof of information consistency for GP models. We next move to our main result stating that we can use the MDL score for GP models for causal discovery.

## A.2 Proof of Theorem 3.1

We assume a causal model with changing mechanisms as in Assumption 1. We also recall Assumptions 2, 3, and 4. The first one ensures that our functional models are well-specified.

**Assumption 2** (Sufficient capacity): For each $i$ there exist functions $f_i^{(\pi)} \in \mathcal{H}_\kappa$ such that Eq. (5) holds.

We also make the following assumption to avoid the pathological case that causal variables $\mathrm{pa}_i^G$ do not affect $X_i$.

**Assumption 3** (Causal minimality): No $f_i^{(\pi)}$ is constant with respect to any of its inputs.

Similarly, we also ensure that interventions have an effect on variables or, put differently, that a mechanism change of $f^{(\pi)}$ indeed corresponds to a difference in distribution of the intervened contexts in $\pi$.

**Assumption 4** ($\Pi^*$-faithfulness): For each $\pi \neq \pi' \in \widehat{\Pi}_i$ we have $f_i^{(\pi)} \neq f_i^{(\pi')}$.

With these assumptions, we are ready to prove our Theorem 3.1, which we repeat here for ease of access too.

**Theorem 3.1** (Identification of MECs)**:** *Let Assumptions 1-4 hold and let $\lambda$ be sufficiently small. Then the estimates $\widehat{G}, \widehat{\mathbf{\Pi}}$ minimizing our score L,*

$$\widehat{G}, \widehat{\mathbf{\Pi}} = \underset{G,\mathbf{\Pi}}{\arg\min}\, L(G, \mathbf{\Pi})\ ,$$

*recover a DAG $\widehat{G} \sim G^*$ in the Markov Equivalence class of the true DAG $G^*$, as well as the true partitions $\Pi_i^*$ for each node $i$ with probability 1*

$$P(\widehat{G} \sim G^*, \widehat{\mathbf{\Pi}} = \mathbf{\Pi}^*) = 1$$

*Proof.* The idea of our proof is similar to those in Yang et al. [36] and Brouillard et al. [9]. We start by assuming that the correct partition $\Pi^*$ is known. Let $\widehat{G}$ be the graph recovered by minimizing $L$ as above. We can write the difference in scores as

$$L(X;\widehat{G}) - L(X;G^*) = \sum_i \sum_{\pi \in \Pi_i^*} D_{\mathrm{KL}}(P(X_i \mid \mathrm{pa}_i^{G^*}, c \in \pi); P(X_i \mid \mathrm{pa}_i^{\widehat{G}}, c \in \pi)) + \Big( L(\widehat{G}) - L(G^*) \Big),$$
(9)

where the dependence on $\Pi^*$ and the functions $f_i^{(\pi)}$ is left implicit.

Assume that $\widehat{G} \subsetneq G^*$. Then at least one of the $D_{\mathrm{KL}}$ terms is non-zero, say $\eta > 0$. However, since $L(G), L(G^*) \propto \lambda$, we can find a $\lambda > 0$ small enough such that $\left| L(\widehat{G}) - L(G^*) \right| < \eta$, and therefore $L(X;\widehat{G}) > L(X;G^*)$.

Now assume that $G^* \subsetneq \widehat{G}$. Then in particular there exists an $i$ for which $\mathrm{pa}_i^{G^*} \subsetneq \mathrm{pa}_i^G$. However, from the causal Markov condition we know that $X_i$ is independent of all its non-descendants given $\mathrm{pa}_i^{G^*}$. In particular, the KL divergence is zero. However, then either $L(\widehat{G}) > L(G^*)$ or $\widehat{f_i^{(\pi)}}$ would be constant with respect to one of its inputs.

From these two cases, it therefore follows that $\widehat{G}$ is Markov equivalent to $G^*$.

For the case where $\Pi^*$ is not known, let $\widehat{\mathbf{\Pi}}$ be the set of recovered partitions for each $X_i$. Then if $\widehat{\mathbf{\Pi}} \neq \mathbf{\Pi}^*$ there must be at least one $i$ for which $\widehat{\Pi}_i \neq \Pi_i^*$. There are two ways in which the partitions can differ.

First, the partition $\widehat{\Pi}_i$ is more fine-grained than $\Pi_i^*$. That is, for each $\pi^* \in \Pi_i^*$ there exist groups $\pi_1, ..., \pi_k \in \widehat{\Pi}_i$ such that $\pi^* = \pi_1 \cup ... \cup \pi_k$ and further there exists at least one $\pi^*$ for which $k > 1$. For this $\pi^*$ and associated $\pi_1, ..., \pi_k$ we have as before that for sufficiently small $\lambda$ the minimizer for $L(X;\widehat{G})$ would have to be $f_i^{\pi_1} = ... = f_i^{\pi_k}$. However, by assumption (A3) this is not permitted. This case therefore cannot occur.

Second, there could be a group $\pi \in \widehat{\Pi}_i$ which intersects at least two different groups $\pi_1^* \neq \pi_2^* \in \Pi_i^*$. However, from (A3) we know that $f_i^{\pi_1^*} \neq f_i^{\pi_2^*}$ so that whatever function $f_i^\pi$ we pick in the discovered model, it would necessarily lead to the wrong distribution for at least one of the two groups and therefore a positive $D_{\mathrm{KL}}$ term, which as before is sub-optimal for $\lambda$ small enough. $\square$

### A.3   Proof of Theorem 3.2

For the proof of Theorem 3.2, we require the following formalization of independent changes of mechanisms.

**Assumption 5** (Independent Mechanism Shift)**:** For all $i \neq j : \widehat{\Pi}_i(C) \perp\!\!\!\perp \widehat{\Pi}_j(C)$.

Under this assumption, we can prove that we can identify the full causal model.

**Theorem 3.2** (Identifiability from independent changes)**:** *Let Assumptions 1-5 hold. Then for $\lambda$ sufficiently small, where $\lambda$ is a hyperparameter bounding the score differences of $\widehat{G}, G^*$, we can identify the causes $\mathrm{pa}_i^G$ and partition $\Pi_i$ of each $X_i$ in the causal model $(G^*, \mathbf{\Pi}^*)$ with probability 1,*

$$\lim_{|\mathcal{C}| \to \infty} P(\widehat{G} = G^*, \widehat{\mathbf{\Pi}} = \mathbf{\Pi}^*) = 1$$

*In particular, the fully directed network $G^*$ is identifiable.*

*Proof.* As we have seen from Theorem 3.1, we can identify the correct MEC and $\mathbf{\Pi}^*$ already. Our goal is therefore to show that any other $\widehat{G}$ in the same MEC as $G^*$ will correspond to a $\widehat{\mathbf{\Pi}}$ different from $\mathbf{\Pi}^*$.

Let us consider the target variable $X_i$. We can ask ourselves what happens when an edge $X_i \to X_j$ is misdirected in $\widehat{G}$. We will show that the misdirected edge will lead to a wrong partition with probability 1 as $|C| \to \infty$.

To that end, note that when $\Pi_i^*(c) = \Pi_i^*(c')$ but $\Pi_j^*(c) \neq \Pi_j^*(c')$ for two different contexts $c, c'$ then in the alternative graph $\widehat{G}$, the distribution $P(X_i \mid \mathrm{pa}_i^{\widehat{G}}, C = c) \neq P(X_i \mid \mathrm{pa}_i^{\widehat{G}}, C = c')$ is not invariant. Therefore, for each set $\pi \in \widehat{\Pi}_i$, the number of functions required to model the distribution of $X_i$ increases. That is, we require one function for each $\widehat{\Pi}_j(c)$ such that $\widehat{\Pi}_i(c) = \pi$.

Now, due to $\widehat{\Pi}_i(C) \perp\!\!\!\perp \widehat{\Pi}_j(C)$ this is guaranteed to be one function for each pair $(\pi, \pi') \in \widehat{\Pi}_i \times \widehat{\Pi}_j$ as $|\mathcal{C}| \to \infty$.

However, this would lead to the *wrong* partitions $\widehat{\Pi}_j \neq \Pi_j^*$, in contradiction to Theorem 3.1 above.

$\square$

## A.4  Proof of Theorem 3.3

We now turn to the probabilistic version of the previous result, for which we need the following assumption.

**Assumption 6** (Fixed Partition Sizes)**:** For each $X_i$, there exists a finite number of sets $B_{i,1}, ..., B_{i,k_i}$ such that $p_{ij} := P(C \in B_{i,j}) > 0$.

With this, we can prove Theorem 3.3.

**Theorem 3.3** (Identifiability in the finite context setting)**:** *Let the Assumptions of Theorem 3.2 as well as Assumption 6 hold. Let $p = \min_{i,j} p_{ij}$ be the minimum probability for a context to fall into one of the bins $B_{i,j}$. Then if $|\mathcal{C}| \geq -2p^{-2}\log(p) > 0$ we have*

$$P(\widehat{G} = G^*, \widehat{\mathbf{\Pi}} = \mathbf{\Pi}^*) = \Omega\left(\exp\left(-|E|\, e^{-c}\right)\right),$$

*where $c = p^2\,|\mathcal{C}| + 2\log p \geq 0$ and $|E|$ is the number of edges in $G^*$.*

*Proof.* The proof is based on the same idea as the proof of Theorem 3.2, and uses probabilistic bounds on the coupon collector problem to obtain the desired result.

That is, we derive bounds on the probability that for an edge $X_i \to X_j$ in $G^*$, there exist pairs of contexts $c, c'$ such that $\Pi_i^*(c) = \Pi_i^*(c')$ and $\Pi_j^*(c) \neq \Pi_j^*(c')$ for all possible combinations in $\widehat{\Pi}_i \times \widehat{\Pi}_j$. We can consider these as bins with probabilities $\geq p^2$, so that for the stopping time $T = |C|_{\mathrm{fin}}$ at which each such bin has been hit at least once we have

$$P\left(T < -2p^{-2}\log(p) + c/p\right) \longrightarrow \exp(-e^{-c}).$$

By using $c = p^2\,|\mathcal{C}| + 2\log p \geq 0$, we therefore have

$$P(X_i \to X_j \in \widehat{G}, \widehat{\Pi}_i = \Pi_i^*) = \Omega\left(\exp\left(-e^{-c}\right)\right),$$

and by taking the naive lower bound of treating all edges independently, we therefore obtain

$$P(\widehat{G} = G^*, \widehat{\mathbf{\Pi}} = \mathbf{\Pi}^*) = \Omega\left(\exp\left(-|E|\, e^{-c}\right)\right).$$

$\square$

## A.5   Discussion of Identifiability Results

Here, we place our identifiability results in the context of related work. In doing so, we explain our observation that our approach already discovers causal directions given a *single* context, as well as state the implications of Thm. 3.3 on *how many* contexts we need to identify the causal DAG.

**Identifiability from a single context**   The work of Peters et al. [45] shows that causal DAGs are fully identifiable from i.i.d. data using *independence of residuals*. These results are obtained by assuming differentiability of the underlying distributions as well as causal mechanisms. By modeling our causal mechanisms via GPs, these assumptions are not applicable since individual GP paths are in general not differentiable. For our more general setting, we are not aware of a proof that the DAG is identifiable from a single context. However, Marx and Vreeken [25] show that in the bivariate case, MDL-based score functions allow to identify the causal direction from i.i.d. data using the independent noise asymmetry. Our empirical results support this as our approach can already direct a reasonable number of edges given a single context as well as in the bivariate case.

**Identifiability depending on the number of contexts**   We remark that our results address the case where our data comes from random environments over which we have no control. When we do have control over the environments, i.e., when we can actively perform interventions with known effects, then general results on the numbers of required contexts apply, requiring data from $\mathcal{O}(\log m)$ environments for $m$ variables [32].

As a second note, in Theorem 3.3, we show how many contexts it takes us to identify both the causal graph *as well as* the partition of the causal mechanisms across environments. If we care only about the identifiability of the causal graph, then two mechanisms per variable (for those variables that are not already directed by Meek's rules) suffice, which dramatically reduces the number of contexts required. In particular, if we obtain samples with equal probabilities for each mechanism, then we require only 20 contexts to orient up to 10000 edges with probability >0.99, independently of the precise structure of the graph. This compares to (roughly) 8 contexts that are required under optimal intervention design for causal learning [32]. Our results however do not require us to have control over the intervention targets, which is realistic in many practical applications where we may encounter unknown distribution shifts or off-target intervention effects [7, 17].

## A.6   Discussion of Assumptions

To conclude our theoretical analysis and point out the limitations of our method, we revisit the main assumptions that we make and discuss in which settings we consider them reasonable.

**Functional Model (Assumption 1,2)**   While we avoid some issues related to conditional independence testing in causal discovery [14, 15], this comes at the cost that we need to make additional assumptions on the functional form of causal mechanisms. As we noted in the main text, Gaussian processes constitute a rich function class and allow for a fully non-parametric approach. While they are more general than the functional model assumed in Peters et al. [45], a similar kernelized scoring function is proposed by Huang et al. [15] which also allows for regularization, so a use of this functional model in a non-i.i.d. setting could be an interesting comparison in future work.

**Invariant Noise (Assumption 1,2)**   By assuming that the noise $N_i$ remains invariant across environments, we follow how related work defines environments [13, 9, 7]. This assumes that different environments arise from changes in the functional dependencies between the variables, commonly known as soft interventions. That is, environments differ due to changes of the conditional $P(X_i \mid \mathrm{pa}_i^G)$ of some variables $X_i$ while the exogenous noise variables $N_i$ remain invariant. Example applications where this holds include knockout or knockdown interventions in genomics, where the expression of a certain gene is prevented or inhibited [16, 17], as well as the molecular interventions in our real-world dataset, where activating or inhibiting reagents are added to the system [20].

To give an example where noise invariance is violated, we consider direct interventions on the noise $N_i$ in our experiments (Fig. 3(a)). Our results suggest that this particular model misspecification is not a limitation in practice. The same is true for perfect intervention (Fig. 3(a)).

**Sufficiency**   There is one other reason why noise invariance may be violated which relates to another assumption, namely causal sufficiency. If there is an unobserved variable, it may introduce a correlation of the noise $N_i$ of different observed variables. Considering contexts allows to address confounding in a limited sense, since the categorical context variable $C$ can represent so-called "pseudo-confounders", such as background stimuli that remain fixed in a given context [4]. However, these need to be a function of the domain index. There may also be a global latent confounder that introduces a correlation between the noises $N^{(c)}$ across different environments. We are not aware of approaches that address this situation, and extending our theoretical framework to general latent confounding is an interesting option for future work.

**Faithfulness (Assumption 3,4)**   Faithfulness assumptions are standard in causal discovery, and in our case assert that the causal parents of a variable indeed affect the variable, as well as that our partitions faithfully represent all changes in causal mechanisms. Both criteria are necessary to detect parents, respectively mechanism changes, from the observed distributions.

**Independent Mechanism Shift (Assumption 5)**   Rather than assuming sparse mechanism shifts, we consider independent mechanism shifts in our work. To formally assume this, we consider our partitions $\Pi$ as independent random variables, which can be thought of as deterministic functions of the context variable $C$. The result is a categorical random variable that indicates the group membership, rather than context membership, of samples. By assuming independence of these random variables and including them in our model, we ensure that incorrectly directed edges in $G$ lead to larger numbers of modeled mechanisms and thus larger values of our score $L$.

To explain this, we recall the example in Fig. 1. The true partition for $Y$ has two groups, and the partition of its causal child $Z$ has either two (Fig. 1(b)) or five (Fig. 1(c)) groups. In the true causal model, the partitions for $Y$ and $Z$ are independent. Indeed, in the causal direction, we see no differences in our MDL scores between both cases (green curves in Fig. 1(b), 1(c)) and discover the true partition for $Y$ with $|f| = 2$ (green star) *independently* of the changing partition for $Z$ in these cases. If we consider an anticausal function $Z \rightarrow Y$, in contrast, there is a difference between both cases (red curves). The partition we discover for $Y$ here *depends* on the partition of $Z$, and in both cases it has more groups (minima of the red curves, $|f| = 4, |f| = 5$) and larger score values compared to the causal direction. In the proof of Theorem 3.2, we show more generally that under independence of the true partitions, incorrectly directed edges will lead to more groups in the discovered partitions as well as larger score values of the model.

We justify the independent mechanism shift assumption in the context of interventional experiments. Consider a gene knockout intervention. This is an exogenous influence and independent of the causal interactions between other genes. Similarly, Sachs et al. [20] perform soft interventions by adding compounds that activate or inhibit specific molecules. In both examples, a mechanism change is due to an external influence and does not depend on the causal mechanism of other variables. In general, this independence holds in generic situations where there is no confounding behind cause and effect [1, 4].

**Fixed Partition Sizes (Assumption 6)**   By fixing partition sizes, we fix the number of different causal mechanisms for each of the variables. In effect, we assume that all heterogeneity in causal effects is captured in our model. For example, if we studied the differences between treatments in different hospitals, then as we obtain more data, these data would be obtained from the same set of hospitals, rather than by the addition of new hospitals. By ensuring that the partition sizes are fixed, we ensure that the things we are trying to learn do not change as we obtain more data. This allows us to give our more practical identifiability result in Theorem 3.3 for a smaller number of contexts.

## B   Algorithms

In this section, we describe the implementation details of the LINC algorithm and its variants.

### B.1   LINC for Learning causal models under Independent Changes

Our approach LINC considers data over continuous variables $X^{(c)}$ in multiple contexts $c \in \mathcal{C}$, and returns a causal graph $G$ and a set of partitions $\Pi$.

---

**Algorithm 1:** LINC

> **Input:** data $X^{(c)}$, candidate DAGs $\mathbf{G}$.
> **Output:** DAG $G$ and partitions $\mathbf{\Pi}$.
> **for** each DAG $G \in \mathbf{G}$ **do**
>     **for** each variable $X_i$ with causes $X_S$ in $G$ **do**
>         **for** each partition $\Pi$ of $C$ **do**
>             **for** each group $\pi$ in $\Pi$ **do**
>                 Pool the data in $\pi$ to obtain $X^{(\pi)}$
>                 Fit a GP $f \sim \mathcal{GP}(X_S^{(\pi)} \to X_i^{(\pi)})$
>                 Compute $L(X_S^{(\pi)}, X_i^{(\pi)} \mid \mathcal{H}_\kappa)$ using Eq. (6)
>             **end for**
>         **end for**
>     **end for**
>     Find the partitions $\widehat{\mathbf{\Pi}_G}$ minimizing Eq. (10)
> **end for**
> Find the DAG $\widehat{G}$ minimizing Eq. (11)
> **return** $\widehat{G}, \widehat{\mathbf{\Pi}_G}$

---

In Alg. 1, we show the exact search algorithm that we consider in the main paper. It traverses a set of candidate causal graphs $\mathbf{G}$. For each graph $G$ and variable $X_i$, it discovers a mixture of mechanisms that generate $X_i$ from its parents in $G$. To do this, we consider each partition of the contexts, fit a GP regression model to the data pooled in each group in the partition, and compute the MDL score for these GPs. We choose the partition that results in the smallest overall description length,

$$
\begin{aligned}
\widehat{\Pi}_i &= \arg\min_\Pi L(\Pi) \\
&= \arg\min_\Pi \sum_{\pi \in \Pi} L(X_S^{(\pi)}, X_i^{(\pi)} \mid \mathcal{H}_\kappa)
\end{aligned}
\tag{10}
$$

where the set $X_S$ corresponds to the parents of $X_i$ in $G$. Repeating this for each variable, we obtain the partitions $\widehat{\mathbf{\Pi}_G}$ for $G$. We return the DAG $\widehat{G}$ with smallest overall description length,

$$
\widehat{G} = \arg\min_{G \in \mathbf{G}} \sum_{\Pi \in \widehat{\mathbf{\Pi}_G}} L(\Pi) .
\tag{11}
$$

Regarding the complexity of our algorithm, we need to address three aspects: the search space over DAGs $\mathbf{G}$, the search space over partitions $\mathbf{\Pi}$, and the GP regression itself. We propose efficient approaches for each aspect in the following, and conclude by giving a complexity analysis.

## B.2 LINC with Greedy Equivalence Search

We first address the problem of discovering causal DAGs. As the search space over DAG models is super-exponential in the number of variables [33], we limit consideration to the Markov Equivalence class of the true model and set $\mathbf{G} = \mathrm{MEC}(G^*)$. We remark that in case the true MEC is unknown, one can apply LINC using a two-stage approach, first using existing causal discovery approaches for multiple contexts [4, 7, 5, 9] to discover a partially directed causal graph, and subsequently applying LINC to orient the remaining edges.

We can also use our MDL score in a score-based DAG search algorithm, such as Greedy Equivalence Search (GES) [33]. GES uses a scoring criterion that evaluates each causal relationship between a given set of causes $X_S$ and an effect $X_i$. Huang et al. [15] show how to use generalized, nonparametric scoring functions in GES, leading to the Generalized GES algorithm (GGES). Our score can be used as a plug-in estimator within this framework because it is a locally consistent scoring criterion due to information consistency of GPs. We note that our score is not score-equivalent, that is, different DAGs in the same MEC may attain different scores, but Huang et al. [15] show that score local consitency is sufficient to use a scoring function with GGES. While we discover a single DAG $G$ over all contexts in this manner, we can account for non-i.i.d.-ness of the data as well as changes of the causal mechanisms.

---

**Algorithm 2: LINC$_{\text{CLUS}}$**

**Input:** data $X^{(c)}$, variable $X_i$ with causes $X_S$.
**Output:** partition $\Pi_i$.
**for** each context $c$ in $C$ **do**
    Regression in the context $f^{(c)} \sim \mathcal{GP}(X_S^{(c)} \rightarrow X_i^{(c)})$
    **for** each context $c' \neq c$ and $c'$ already seen **do**
        Compute $L'(X_S, X_i \mid c, c')$ using Eq. (12)
    **end for**
**end for**
Find $\Pi = \arg\min_\Pi \sum_{\pi \in \Pi} \sum_{c,c' \in \pi} L'(X_S, X_i \mid c, c')$ using clustering, e.g. by solving Eq. (13)
**return** $\Pi$

---

We propose exploring combinations of LINC with different DAG search frameworks in future work, and focus on orienting edges within a MEC in our experiments. This is motivated by two reasons. Firstly, it allows to study the behavior of our scoring function in isolation, taking out the effect of greedy DAG search algorithms, such that our experimental evaluation can support our theoretical results. Secondly, our closest competitors are the mechanism counting scores that need to start from a base MEC [12, 13] such that we also evaluate LINC on orienting edges within a MEC.

### B.3 LINC with Clustering

To scale efficiently with the number of contexts, we introduce a variant of LINC that, rather than all set partitions, considers pairwise combinations of contexts.

To this end, we adapt the Minimum Description Length (MDL) score for GPs. We want to measure whether the data $X^{(c)}, X^{(c')}$ over two contexts $c, c'$ is preferably encoded using separate GP functions for each, or a single model over the pooled data. A natural way to compare different modelling hypotheses is via the gain in compression of one model over the other.

By the no-hypercompression inequality [29], the probability that a model other than the optimal one should achieve a substantially better compression for a dataset is negligible. More specifically, the probability that the competing model leads to a gain in compression of $k$ bits if it is *not* the true model is smaller than or equal to $2^{-k}$. For a significance threshold $\alpha$, we can consider a gain of $k$ bits as insignificant in case $2^{-k} > \alpha$ [29].

Hence, we consider the gain $L'$ in compression that two separate GPs achieve over a joint model,

$$
\begin{aligned}
L'(X_S, X_i \mid c, c') = {} & L(X_S^{\{c,c'\}}, X_i^{\{c,c'\}}) \\
& - \left( L\big(X_S^{(c)}, X_i^{(c)}\big) + L\big(X_S^{(c)}, X_i^{(c)}\big) \right).
\end{aligned}
\tag{12}
$$

Given that the gains should not be positive by no-hypercompression, we minimize the pairwise scores $L'$. This can be seen as a graph clustering problem on a graph with nodes $c \in C$, weighted edges between each pair $c, c'$ with weights $L$ taking positive values iff $c, c'$ belong to the same cluster, and where the number of clusters is unknown. Miyauchi et al. (2018) consider exactly this problem and show that it corresponds to the following Integer Linear Program (ILP) objective,

$$
\min \sum_{c_i, c_j} L'(c_i, c_j) x_{ij}
\tag{13}
$$

where $x_{ij}$ are decision variables indicating group membership, $x_{ij} = 1$ if $c_i, c_j \in \pi$ and zero otherwise. The minimization is subject to so-called triangle constraints [53] ensuring that the solution is a valid partition of $\mathcal{C}$, and can be solved using standard solvers.

As another variant, we can use hierarchical clustering with distance measure $L'$. With this approach, we discover the best partition of a given size $k = |\Pi|$. We select the best $k$ by comparing the full MDL scores of the partitions at $k$ and returning the minimal one.

We show how to adapt LINC in Alg. 2, and refer to it as LINC$_{\text{CLUS}}$.

### B.4 LINC with Random Fourier Features

Last, we show how our score can be adapted to a common approximation of GPs using Random Fourier Features (RFFs) [34].

Computing $L$ for GPs is computationally expensive since GP regression does not scale well with the number of samples. One way to mitigate the complexity of GP regression is to express its kernel through a Fourier transform, which in turn can be approximated efficiently. This is known as the Random Fourier Feature (RFF) approximation [34].

The approach is as follows: if $\kappa$ is shift-invariant, we have

$$\kappa(x, y) = \int p(\omega) \cos(\omega^\top (x - y)) d\omega$$

$$\approx \frac{1}{R} \sum\nolimits_{r=1}^{R} \cos(i\omega_r^\top (x - y)) = z(x)^\top z(y)$$

for some probability measure $p(\omega)$. For the RBF kernel, this measure $p(\omega)$ is another Gaussian distribution [18]. Here, $z(x) = \frac{1}{\sqrt{R}}(\sqrt{2}\cos(\omega_1^\top x), ..., \sqrt{2}\cos(\omega_R^\top x))^\top$ and $z(y)$ similarly. Thus, for each pair $x_i, x_j$, we obtain an unbiased estimator

$$\kappa(x_i, x_j) = z(x_i)^\top z(x_j) .$$

The RFF approximation of GP prediction uses the regression $f(x) = z(x)^\top w$. We define our MDL encoding using the log-likelihood of this regression, as well as the norm

$$\|f\|_\kappa^2 = w^\top Z w$$

for the feature matrix $Z = z(X)z(X)^\top$. Up to these modifications, we use the MDL encoding in Eq. 6 for RFF regression models.

### B.5 Complexity

For DAG discovery, exact search is infeasible due to the super-exponential search space over causal DAGs [33]. There are greedy solutions in $\mathcal{O}(|\mathcal{V}|^3)$, $\mathcal{V}$ being the number of nodes in the DAG [26].

Regarding the partition discovery problem, the exact solution involves the evaluation of $b$ partitions, $b$ being a bell number in $|\mathcal{C}|$. Heuristic search requires only $|\mathcal{C}|^2$ regressions. While the problem of finding an optimal clustering of $\mathcal{C}$ (e.g., using the ILP) is still NP-hard [53], there exist efficient options such as $k$-means that we could insert. In addition, we have $|\mathcal{C}| \ll N$, and ILP search can be done using standard solvers, which we found to be efficient in practice.

Lastly, exact GP regression is in $\mathcal{O}(N^c)$ for $N$ data points and $c \approx 2.7$, while the RFF approximation is in $\mathcal{O}(\min\{N^3, M^3\})$ for $M$ features [34].

## C  Experiments

In this section, we include additional information on the experiments we present in the main paper. In addition, we illustrate how LINC and its variants introduced in the previous section scale in practice.

### C.1  Causal Discovery Beyond MEC

First, we revisit our main experiment on causal discovery beyond MEC in Fig. 2(a). As we show F1 scores in this figure, we also include the Precision and Recall in Fig. 5.

We can observe that LINC performs well in terms of edge recall in all settings, while maintaining precision comparably high to MSS. Only for large edge densities, LINC is surpassed by MSS, while we outperform competitors for small DAG sizes, small edge density, and under few mechanism changes respectively for a small number of contexts. In particular, for no mechanism changes ($|S| = 0$ and $|C| = 1$) and when few variables are involved in the causal relationships ($|G| = 3$ and $p = 0.3$), we can see that the recall values are close to 0.5 (dashed lines) suggesting that the DAGs recovered by the competitors are close to the known base MEC, while LINC learns additional causal directions.

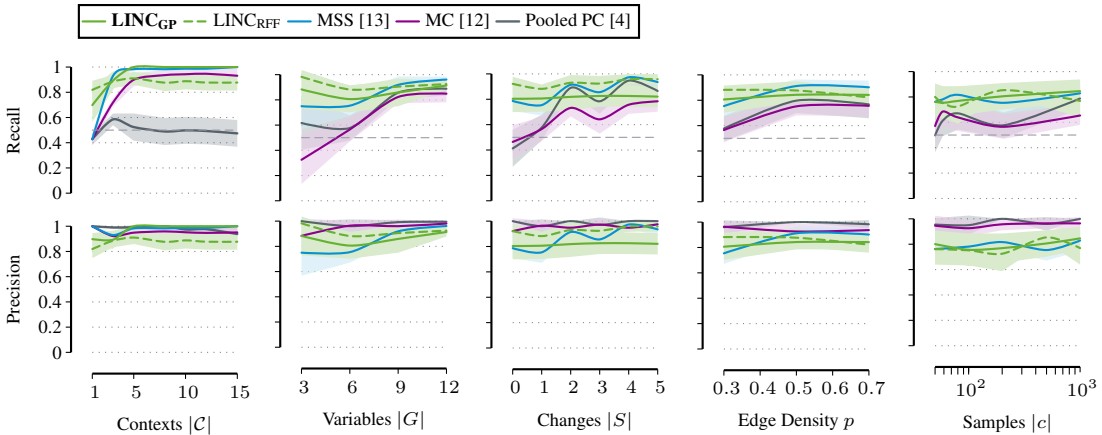

Figure 5: *LINC disovers causal models beyond Markov Equivalence.* We report Recall and Precision on discovering causal edges within a MEC. As in the main paper, we consider synthetic data from different contexts, with the base settings of $|\mathcal{C}| = 5$ contexts, $|G| = 6$ variables, $|S| = 2$ intervened variables, edge density $p = 0.3$, and $|c| = 500$ samples.

## C.2 Scalability Experiments

Given that we suggested more efficient variants of LINC, we evaluate their performance and scalability here. We are interested in whether $LINC_{CLUS}$ can efficiently discover partitions for increasing $|\mathcal{C}|$, whether $LINC_{RFF}$ improves runtimes for increasing sample sizes $|c|$, and whether our method and competitors are applicable to larger DAG sizes $|G|$.

Throughout this section, we consider the main experimental setup of discovering causal models beyond Markov Equivalence, and use the settings $|S| = 1, \mathcal{C} = 3, |c| = 500$ and $|G| = 3, p = 0.3$.

### C.2.1 LINC with Clustering

In Fig. 6, we compare the variant of LINC that exhaustively enumerates all context partitions (Alg. 1) to $LINC_{CLUS}$ which relies on clustering the contexts by compression gain (Alg. 2). As the clustering procedure in $LINC_{CLUS}$, we consider Integer Linear Programming (ILP) as well as agglomerative clustering (aggl.).

As an exhaustive search over all partitions did not finish running for $|\mathcal{C}| = 20$ in a reasonable time (one day), we include partial results (gray).

For simplicity, we instantiated LINC with RFFs in all cases. While an enumeration of all partitions of the contexts is not feasible for more than 20 contexts, we found the clustering approaches to scale easily to at least 50 contexts. In particular, both result in high F1 scores for directing edges in a MEC given enough contexts, with perfect F1 scores when given more than 15 contexts. This suggests that in practice, rather than computing the full MDL score for each partition, we can consider using pairwise MDL compression gains to apply LINC to many contexts.

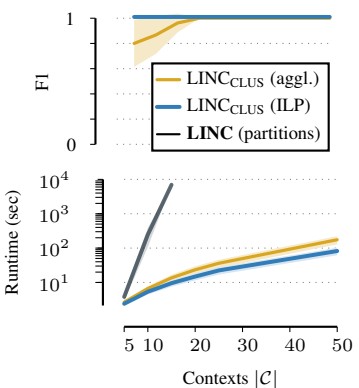

Figure 6: *LINC can be combined with clustering to discover partitions.* We compare exhaustive enumeration of partitions (gray, Alg. 1) to clustering (Alg. 2).

### C.2.2 LINC with Random Fourier Features

Next, we turn to the kernel approximation through Random Fourier Features (RFFs) described in the previous section.

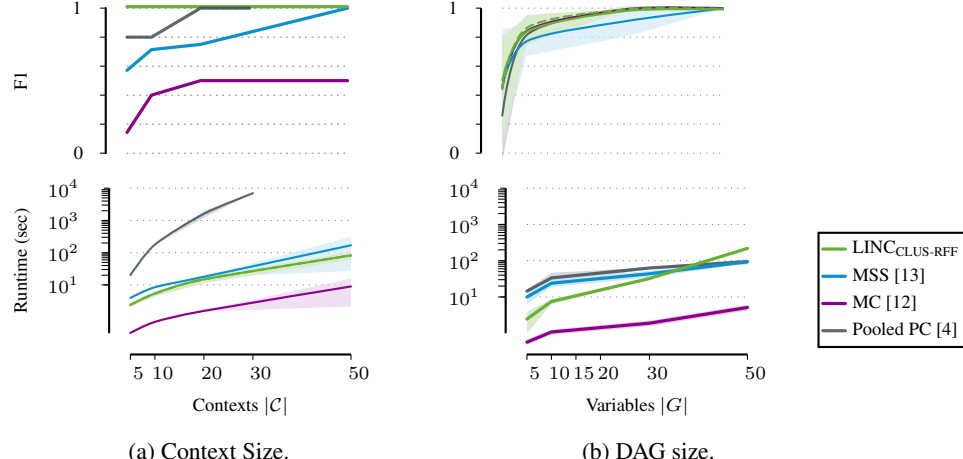

(a) Context Size.  (b) DAG size.

Figure 8: *LINCs variants are accurate while being scalable.* We compare LINC with RFFs and clustering to the baselines, for (a) an increasing numbers of contexts, and (b) increasing DAG size.

We also included RFFs in our main evaluation in Fig. 5 to confirm that they work well in practice. To illustrate the computational benefit, we also compare the runtimes of GPs and RFFs in Fig. 7. We again use the settings $|S| = 1$, $|c| = 500$ and $p = 0.3$ and consider $|C| = 3$. We confirm that RFFs mitigate the sample complexity of the GP regression without notably affecting performance.

### C.2.3 LINC and Competitors

To conclude, we compare the efficient variants of LINC to its competitors. We are interested in whether the clustering variant of LINC combined with RFFs is scalable while being accurate.

In Fig. 8, we show our experiment on discovering causal edges for up to 50 contexts, resp. variables. As we found no noticeable difference between the clustering approaches that we instantiate LINC with, we show only the ILP version to avoid clutter.

LINC scales to at least 50 contexts, while Pooled PC was not applicable to more than 30 contexts in a reasonable time, even on 3-variable DAGs (Fig. 8(a)). We also find that LINC scales to DAGs with at least 50 variables with comparable runtimes to

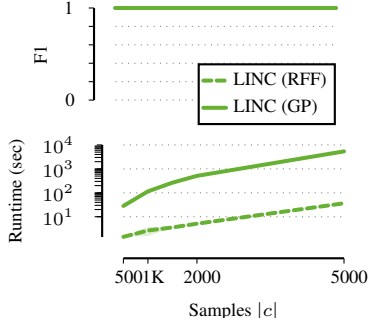

Figure 7: *LINC can be combined with RFFs to improve sample complexity.* We compare LINC with the GP MDL score (solid) to the RFF approximation (dashed).

the nonlinear baselines (Fig. 8(b)). Regarding performance, in Fig. 8(a) we can see that MC discovers only the base MEC due to the small number of variables. Pooled PC and particularly MSS need a larger number of contexts in this case, whereas the efficient version of LINC discovers all causal directions given five contexts. We conclude from our experiments that the efficient versions of LINC reliably perform in practice.

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
