# OpenReview forum: "Learning Causal Models under Independent Changes"
_NeurIPS.cc/2023/Conference — NeurIPS 2023 poster_

### Official Review · Reviewer_NaRE · 2023-07-05

**Soundness:** 3 good
**Presentation:** 2 fair
**Contribution:** 2 fair
**Rating:** 7
**Confidence:** 3

**Summary:**

The author propose a score-based method for causal discovery from multi-environment data. Identifiability of the causal model and the environment partition were shown for the proposed score function. The proposed algorithm was evaluated on synthetic and multiple real data sets.

**Strengths:**

1. The toy example provides a nice illustration of the proposed idea.

2. Using a measure of the complexity of GP models as the score function for causal models is an interesting idea.

3. There are sufficient evaluations on synthetic and real data sets.

**Weaknesses:**

1. The score function (4) is written in terms of the true conditional probability, while the empirical score function is not defined. In the algorithm, it was not mentioned how to estimate the score function using the data.

2. There are missing explanations and plenty of typos in the main results.

Missing explanations:

(1) The role of the the penalty term $R(X_{S})$ in $(4)$ should be explained even it is from a previous work.

(2) The term $L(h)$ in $L(X|\mathcal{H})$ is not defined in Section 2.2.

(3) How is $\mathcal{H}_{k}$ constructed in the algorithm? How large is the function class?

Typos:

(1) Whether the term $-log P(X_{i}|X_{S})$ in (4) should be $-log P(X_{i}|f(X_{S}))$ or something else?

(2) LHS of (5) should be $X_{j}^{(c)}$.

(3) $\lambda$ in Theorem 3.2. is not defined in the score function.

(4) "Theorem3.1 assumes Assumptions 1-4, Theorem 3.2 assumes Assumptions 1-5, and Theorem 3.3 additionally assumes Assumptions 6."

(5) $X_{S} \subseteq X\setminus X_{i}$ in (4).


**Questions:**

Overall, I think the idea is interesting and promising, but the authors should check for typos carefully.

**Limitations:**

No potential negative societal impact.

---

> ### Author Rebuttal · Authors · 2023-08-09
>
> We thank you for your time and comments, please find our responses below.
>
> **Score function**
> > The score function (4) is written in terms of the true conditional probability, while the empirical score function is not defined. In the algorithm, it was not mentioned how to estimate the score function using the data. [...]
> > How is $H_\kappa$ constructed in the algorithm? How large is the function class?
>
> In practice, we fit a GP regression from the conditioning set to the target variable. Hence, we do not explicitly construct the hypothesis space $H_\kappa$. As we noted in Section 3.1, since $\kappa$ is the RBF kernel, $H_\kappa$ is dense in the set of continuous functions. However, the beauty of GP models is that despite this it is simple to learn the optimal model from any finite sample.
> As it is fairly standard in GP regression we omitted details on how to estimate the score function, but will make this more clear.
>
> **Missing explanations**
>
> > (1) The role of the the penalty term $R(X_S)$ in (4) should be explained even it is from a previous work.
>
> The penalty is known as the regret term. For the regret of GP models, Kakade et al. (2005) give worst-case bounds [1]. We will add an explanation to our manuscript, thank you for pointing this out.
>
> > The term L(h) in L(X|H) is not defined in Section 2.2.
>
> That is a good point, $L(h)$ refers to a length function or code for the model $h$, and its definition depends on the given model class. In section 3.2, it is given by $||f||_{\kappa}^{2}$.
>
> > Overall, I think the idea is interesting and promising, but the authors should check for typos carefully.
>
> Thank you for listing out typos, we will fix these in our revision.
>
> Please let us know whether we could address your concerns and whether you have further questions.
>
> **References**
>
> [1] Kakade, Sham M., Matthias W. Seeger and Dean Phillips Foster. "Worst-Case Bounds for Gaussian Process Models." *NIPS (2005)*.

---

> > ### Comment · Reviewer_NaRE · 2023-08-17
> > **Reply to rebuttle**
> >
> > Thanks for the clarifications of the GP model.
> >
> > Again, I think the main problem of the paper is the writing. Especially the missing details. I vote for acceptance and adjust my score accordingly.

---

> > > ### Author Response · Authors · 2023-08-18
> > >
> > > Thank you again for your feedback, we will include the missing details that you pointed out in our manuscript.

---

### Official Review · Reviewer_oMbi · 2023-07-05

**Soundness:** 3 good
**Presentation:** 2 fair
**Contribution:** 3 good
**Rating:** 7
**Confidence:** 4

**Summary:**

The authors present a novel approach for causal discovery that goes beyond partially directed graphs by utilizing Gaussian Process (GP) models. The proposed method aims to identify
the correct causal model under certain conditions. The key idea is to leverage algorithmic independence to achieve a concise and lossless description of the data, particularly in the presence of multiple contexts. Unlike existing approaches such as Sparse Mechanism Shift (SMS), which require conditional independence tests, the proposed method employs a scoring criterion based on GP models.
The authors established the theoretical soundness of this approach by providing a clear and concise explanation of the underlying principles. Additionally, they demonstrate the effectiveness of the method through several examples, which serve to evaluate its performance and help to illustrate the practical implications and potential benefits of the proposed method. Overall, the proposed method offers a new perspective on causal discovery by utilizing Gaussian Process models and algorithmic independence.


**Strengths:**

LINC (Learning causal models under Independent Changes) proposed in this paper seems original utilizing GP and its complexity measure for discovering a mixture of mechanisms. Non-iid data can be typically challenging but in this paper’s setting, multi-context is a key to identifying the underlying causal structure. The overall flow of the paper is smooth, and it appropriately makes necessary assumptions.

**Weaknesses:**

Not a major weakness but minor weaknesses (more like comments)

Readability: It would have been beneficial to include explanatory information on prerequisite knowledge to enhance the accessibility of the paper. In particular, “algorithmic independence” seems very crucial, but it is just referenced without properly articulating its definition or the difference to conditional independence.

Assumptions numbering is problematic. It seems that the authors removed one of the assumptions at the very last stage of writing. E.g. Assumption 5 is mentioned in Theorem 3.1, which is in the previous section. Also for Theorem 3.2 refers Assumption 6. Theorem 3.3 calls for Assumption 7 which does not exist.

It is less clear whether C is part of data or not.


**Questions:**

In Theorem 3.1, lambda seems a bit abrupt. Is this the hyperparameter for RBF?

Instead of presenting Theorem 3.2 before 3.3, you may introduce 3.3 and just make 3.2 corollary of 3.3 where |C| goes infinity. (or does 3.3 make an additional assumption (e.g. fixed partition size) which is not required for 3.2?)

I am not entirely sure whether the expression in Assumption 5 is valid. In the following paragraph the authors mentioned that partitions are random variables. Does it mean that C is a random variable where \Pi is just a deterministic function wrapping C?

In Figure 1(b), Lines 40-41 mention that X and Z are changed. But the Figure 1(a) shows the change wrt context is the function of Y not X nor Z. What am I missing?

In Sec 3.5 I couldn’t parse “Given the limited number of distinct context distributions … “ I couldn’t see any P( C ) defined clearly.


**Limitations:**

It seems that the critical part of the paper is Section 3.5 where the authors actually operationalize theoretical results in the previous sections. But I felt that the authors somehow hide some issues? E.g., finding partitions is exhaustive (or equivalently, exponential in the number of contexts)  (despite line 227 mentions that the time complexity  is in supplementary material…)

---

> ### Author Rebuttal · Authors · 2023-08-09
>
> We thank you for your comments, which we address in the following.
>
> **Limitations**
> > It seems that the critical part of the paper is Section 3.5 where the authors actually operationalize theoretical results in the previous sections. But I felt that the authors somehow hide some issues? E.g., finding partitions is exhaustive (or equivalently, exponential in the number of contexts) (despite line 227 mentions that the time complexity is in supplementary material…)
>
> In the main paper we only consider the exact version of LINC, which searches for the optimal partitioning by exhaustively considering all possible partitions. This provides guarantees on the output, at the cost of an indeed exponentially sized search space. As the number of contexts in real-world datasets is typically fairly small, e.g. less than ten contexts for the data by Sachs et al. (2005), we consider the exact version of LINC applicable in practice.
>
> In the Appendix, we additionally provide a faster, heuristic version of LINC that employs clustering to find a good partitioning of the contexts. Although it does not come with theoretical guarantees, it does performs equally well in practice and does so much faster. As we are primarily interested in the guarantees, we consider this version of LINC as a 'fun' additional result, rather than a main contribution.
>
> **Clarity**
> > In particular, “algorithmic independence” seems very crucial, but it is just referenced without properly articulating its definition or the difference to conditional independence.
>
> That is a good point, we will include the definition in the paper.
> In effect, as noted in Postulate 2.1, the algorithmic independence of causal mechanisms amounts to the idea that compressing them independently of one another is optimal.
>
> > It is less clear whether C is part of data or not.
>
> $C$ is indeed part of the data, that is, we need to know in advance which context the data samples come from.
>
> > In Theorem 3.1, lambda seems a bit abrupt. Is this the hyperparameter for RBF?
>
> You are right, and we will introduce $\lambda$ more clearly. It is a hyperparameter that we need for proof to bound the score differences between the true DAG and the one minimizing our score.
>
> > Instead of presenting Theorem 3.2 before 3.3, you may introduce 3.3 and just make 3.2 corollary of 3.3 where |C| goes infinity. (or does 3.3 make an additional assumption (e.g. fixed partition size) which is not required for 3.2?)
>
> Yes, Thm. 3.3. makes the assumption of fixed partition sizes which is not required for Thm. 3.2, which is why we presented them in this order. We will fix our assumption numbering to make this clear.
>
> >I am not entirely sure whether the expression in Assumption 5 is valid. In the following paragraph the authors mentioned that partitions are random variables. Does it mean that C is a random variable where \Pi is just a deterministic function wrapping C?
>
> Exactly, you can see $\Pi$ as deterministic functions of C. The result is a categorical random variable that indicates the group membership, rather than context membership, of samples. While this is a slight abuse of notation, we want to consider $\Pi$ as random variables here so that we can assume their independence.
>
> >In Figure 1(b), Lines 40-41 mention that X and Z are changed. But the Figure 1(a) shows the change wrt context is the function of Y not X nor Z. What am I missing?
>
> As you say, the function of $Y$ changes in both examples, but in addition, there are causal functions of $X$ and $Z$ that also change. In Fig. 1(b), we show what happens when $X, Z$ both have two causal functions, resp. five functions in Fig. 1(c). We compare these two situations to show that the true causal functions for $Y$ are not affected by this (independent).
>
> In the illustration, we omitted the causal functions for $X, Z$ to avoid clutter, but we can include them to make it more clear.
>
> >In Sec 3.5 I couldn’t parse “Given the limited number of distinct context distributions … “ I couldn’t see any P( C ) defined clearly.
>
> Our writing is unclear here; in general, the number of contexts can be arbitrarily large, but in practical applications, we usually only have few contexts available, for example, we have fewer than ten interventional experiments in the data by Sachs et al. (2005).
>
> We are happy to respond to additional concerns and questions.

---

> > ### Comment · Reviewer_oMbi · 2023-08-12
> >
> > Thanks for the response and I am happy to keep my positive assessment of the paper.

---

> > > ### Author Response · Authors · 2023-08-18
> > >
> > > Thanks for your quick response, we will take your points into account in the revised version.

---

### Official Review · Reviewer_LpAE · 2023-07-08

**Soundness:** 2 fair
**Presentation:** 2 fair
**Contribution:** 3 good
**Rating:** 7
**Confidence:** 3

**Summary:**

This paper addresses the problem of causal discovery with heterogenous data coming from multiple contexts where contexts are characterized by soft/hard interventions. Previous work differs in assumptions on how non-iid data is produced, the primary assumption being the Sparse Mechanism Shift assumption which assumes that the number of mechanism changes is small. The current paper proposes a score-based approach to discovering the casual graph from data coming from different contexts. For each variable, the set of contexts can be partitioned where in each bin of the partition, the mechanism that represents the cause-effect relationship between the variable and its parents is unchanged. The score-based approach comprises of a) functionally modeling the relationship between variables and their parents using Gaussian processes (GPs) for each context bin and b) computing a score that is based on minimum description length (MDL) of the GP model. The paper provides identifiability guarantees for identification up to its Markov Equivalence class (MEC). With an additional assumption about independence of context-partitions the authors are able to obtain asymptotic (in the number of contexts) identification guarantees beyond the MEC. Numerical results on synthetic, semi-synthetic and real datasets validate the proposed method's superiority over existing methods.

**Strengths:**

This paper provides a novel algorithm for an important problem of causal discovery using heterogenous data. The main strength of the paper lies in the experimental validation of its proposed method, LINC. The evaluation is done on multiple types of datasets and LINC is shown to outperform or is at least competitive to existing methods on all. While, the idea of using MDL-based scores is not novel in causal discovery, the idea of combining Kakade et. al.'s work in causal discovery is novel. On the theoretical side, while I have questions about the assumptions that I elaborate in the sections below, the paper proves identifiability guarantees in both the asymptotic (in number of contexts) and and finite-context settings.

**Weaknesses:**

Listing out a few weaknesses:
1) Justification of assumptions: The core result of identifiability beyond the MEC depends on the independent mechanism shift assumption. I wasn't convinced about why this assumption makes sense. In particular, why does the assumption imply Line 184-185. I also didn't find any practical justification of the fixed-partition sizes assumption. In general, justifying the assumptions more clearly and with a practical example in mind can help strengthen this weakness.
2) Writing: The writing can be improved greatly. There are undefined notations and a lot of important content of the paper that has not been explained in the first 9 pages. Some examples: a) The main score function in Line 161 is not explained or defined anywhere. Given that LINC is explained only in the appendix, I found it difficult to even understand what the score function was. b) Independent mechanism shift also contains notation \Pi_i(C) which hasn't been defined before (but is clear from context). c) Assumption numbers mismatched. d) words used without defining - algorithmically independent in line 109, direct intervention in Line 34,
3) Experiments: Most of the evaluation is based on recovery of the causal graph whereas identification guarantees also address recovering the partitions.  Can some experimental results be shown to verify if the partitions are being discovered correctly?

**Questions:**

Overall, I think this is a potentially good submission subject to clarifications on a few questions, some of which I have added in the weaknesses section, and some that I outline below:
1) Algorithm 1 (LINC) has a step where it cycles through candidate graphs in MEC.
2) LINC seems to be doing quite well even with just iid data and also in cases with very few contexts. Any explanation about this phenomenon would be interesting.

**Limitations:**

Authors don't address any potential negative impacts explicitly. Since this is an incremental advance of an established line of research, I don't see an issue.

---

> ### Author Rebuttal · Authors · 2023-08-09
>
> We thank you for your detailed feedback.
>
> **Independent Mechanism Shift Assumption**
> > The core result of identifiability beyond the MEC depends on the independent mechanism shift assumption. I wasn't convinced about why this assumption makes sense.
>
> The assumption is easiest to understand in the context of interventional experiments. Consider a gene knockout intervention. This is an exogenous influence and independent of the causal interactions between other genes. Similarly, Sachs et al. (2005) perform soft interventions by adding compounds that activate or inhibit specific molecules. In both examples, a mechanism change is due to an external influence and does not depend on the causal mechanism of other variables. In general, this independence holds in generic situations where there is no confounding behind cause and effect (Huang et al. 2020).
>
> > In particular, why does the assumption imply Line 184-185.
>
> *Lines 184-185 from the paper:*
> > we consider our partitions $\Pi$ as independent random variables. This ensures that incorrectly directed edges lead to larger numbers of modeled mechanisms and thus larger values of our score $L$.
>
> To see this, consider the example in Fig. 1 of the manuscript. The true partition for $Y$ has two groups, and the partition of its child $Z$ has either two (Fig. 1b) or five (Fig. 1c) groups. In the true causal model, the partitions for $Y$ and $Z$ are independent. Indeed, in the causal direction, we see no differences in our MDL scores between both cases (green curves in Fig. 1b, 1c) and discover the true partition for $Y$ (green star). If we consider an anticausal function $Z \to Y$, in contrast, there is a difference between both cases (red curves). The partition we discover for $Y$ depends on the partition of $Z$, and in both cases it has more groups and larger score values compared to the causal direction.
>
> In the proof of Thm. 3.2, we show more generally that under independence of the true partitions, incorrectly directed edges lead to more groups and larger score values of the discovered partitions.
>
> **Fixed Partition Sizes**
> > I also didn't find any practical justification of the fixed-partition sizes assumption.
>
> By fixing partition sizes, we fix the number of different causal mechanisms for each of the variables $X_i$. In effect, we assume that all heterogeneity in causal effects is captured in our model.
> E.g., If we studied the differences between treatments in different hospitals, as we obtain more data, these data would be obtained from the *same* set of hospitals, rather than by the addition of new hospitals. By ensuring that the partition sizes are fixed, we ensure that the things we are trying to learn do not change as we obtain more data.
>
>  **Experiments**
> > Can some experimental results be shown to verify if the partitions are being discovered correctly?
>
> This is a good point. To illustrate this, we sample causal DAGs with our base settings. We pick a variable at random and evaluate whether LINC discovers the partition that shows its mechanism shifts. To do so, we count the context pairs that LINC correctly assigns to a different group (TP), the same group (TN) or mistakenly to a different (FP) or same group (FN). We report precision, recall, and F1 over 100 runs.
>
> | | Recall | Precision | F1 |
> |-|-|-|-|
> | **Mechanism Shift** |0.88|0.83|**0.85**|
> | Noise Scaling |1.0|0.84|**0.91**|
> | Hard Iv. (synthetic) |0.62|0.62|**0.62**|
> | Hard Iv. (SERGIO) |0.68|0.68|**0.68**|
>
> The first row shows the case where the causal mechanism of our variable changes, and we confirm that LINC discovers a high fraction of mechanism changes (recall) with few false discoveries (precision). We also included the cases where our model is misspecified, where LINC scales especially well to noise intervention.
>
>  **Algorithm**
>  > Algorithm 1 (LINC) has a step where it cycles through candidate graphs in MEC.
>
> As candidate graphs, we start from all DAGs within the Markov Equivalence class of the true DAG. If this class is unknown, there exist standard methods to infer it [1].
>
> **The i.i.d. Case**
> > LINC seems to be doing quite well even with just iid data and also in cases with very few contexts. Any explanation about this phenomenon would be interesting.
>
> This is an interesting observation that can give an insight into the benefits of an MDL-based approach. For the i.i.d. case, we refer to identifiability results by Marx and Vreeken (2019) [2]. The idea is that under nonlinear additive noise models, regression residuals are independent in the causal, but not in the anti-causal direction. The authors show that MDL-based scoring functions can capture this, as residual independence results in a better compression in the causal direction. This explains why our MDL-based score can determine causal directions on i.i.d. data, while the mechanism shift counting scores recover the MEC (Fig. 2a).
>
> As for the case of very few contexts, it is known that under optimal intervention sets, only $O(\log(m))$ contexts suffice to obtain full identifiability for arbitrary graphs, suggesting that in general only few contexts are likely needed. [3] Similarly, from our theoretical results we find that for the case with only two mechanisms per variable occurring with probabilities $p = 0.5$ each, identifiability is highly probable when we observe around 20 contexts even when the ground truth contains 10000 edges.
>
> Please let us know if you have additional concerns.
>
> **References**
>
> [1] Marx, Alexander, and Jilles Vreeken. “Identifiability of Cause and Effect Using Regularized Regression.” In *KDD*, 2019.
>
> [2] Joris M. Mooij, Sara Magliacane, and Tom Claassen. 2020. "Joint causal inference from multiple contexts." *J. Mach. Learn. Res.* 21, 1, Article 99 (January 2020), 108 pages.
>
> [3]  Hauser, Alain, and Peter Bühlmann. "Two optimal strategies for active learning of causal models from interventional data." *International Journal of Approximate Reasoning* 55.4 (2014): 926-939.

---

> > ### Comment · Reviewer_LpAE · 2023-08-17
> > **Response to rebuttal**
> >
> > Thanks for the detailed response. My concerns have been largely addressed. I have updated my score accordingly.

---

> > > ### Author Response · Authors · 2023-08-18
> > >
> > > Thank you for your response. We will update our manuscript accordingly.

---

### Official Review · Reviewer_t48a · 2023-07-25

**Soundness:** 2 fair
**Presentation:** 2 fair
**Contribution:** 2 fair
**Rating:** 4
**Confidence:** 3

**Summary:**

The authors study the problem of causal discovery from data under different conditions/contexts. The approach uses an algorithmic model of causation, where the idea is that causal mechanisms provide short (or simple) descriptions of the observed data. Under this principle, the authors propose a score function for models where the functional mechanisms are Gaussian processes (GPs). The main contributions of this work are Theorems 3.1, 3.2, and 3.3, where the authors show identification of the MEC or ground-truth DAG. The core assumptions for such theorems are:
* Causal sufficiency
* All contexts share the same DAG
* Additive noise model with GPs
* Causal minimality
* $\Pi$-faithfulness
* Independence of mechanism shifts
The authors develop LINC to learn the causal DAG from different contexts, and provide some experiments to validate their results.

**Strengths:**

### Potential Reasons for Acceptance
   - The paper provides a fair investigation into the core assumptions that many causal modeling approaches, including SMS, rely upon, and moves the field forward by addressing these assumptions.
   - Novelty of the proposed approach "LINC": By adopting the algorithmic notion of independence and Gaussian Process models, the authors elevate the ability to identify the accurate causal model and extend the scope beyond partially directed graphs.
   - Identifiability theory: The authors provide theoretical justification for their approach, followed by some empirical evaluations on both synthetic and real-world datasets which aim to validate their claims.


**Weaknesses:**

### Potential Reasons for Rejection
   - The DAG from a single context is identifiable as it is a nonlinear additive noise model.
   - Theorem 3.2 seems very unrealistic when considering infinite number of contexts. While the authors try to provide some justification to it by providing a finite-sample statement in Theorem 3.3, I got confused with the idea of $C$ falling into some "bins". What does it mean for $C$ to fall into two or more different bins? The notion of bins were never used until this theorem and was not properly described  in my opinion.
   - Dependency on number of datasets:
     - The effectiveness of the proposed model heavily depends on certain characteristics of the dataset such as having enough contexts, which might not always be the case.
  - Complexity of the proposed solution:
     - The proposed solution is relatively complex, particularly concerning the computational cost. Several experiments are performed on a very small number of nodes, e.g., six-node graphs.
     - Some heuristics are provided in the appendix to alleviate this issue, however, some details are missing. For example, in  Figure 8(b), MC seems faster and obtains similar F1 scores to LINC.
   - The writing can be improved by a fair margin. Learning a DAG from different contexts is not a new setting and the authors could greatly reduce the amount text of somewhat repetitive definitions of the model in Line 69, Assumption 1 and Assumption 2. Indeed Assumption 1 seems useless if Assumption 2 already holds, no? If the theoretical results rely on GPs, why not simply state that in the problem setting? Finally, Theorems 3.1 and 3.2 mention a  "sufficiently small $\lambda$", but such $\lambda$ does not appear in the result. By looking at the text there is a $\lambda$ from the RBF in Line 139; however, in the appendix I got confused as it seems to refer to a different quantity.

**Questions:**

* Why is $|E|$ in the lower bound in Theorem 3.3. inside exp? I might be missing something but doesn't this come from the union bound and, hence, shouldn't it be outside exp?
* My main criticism of this work is that the model under consideration is identifiable from observation data (Peters et al., 2014). Moreover, since the authors assume that **all contexts share the same DAG**, why couldn't one identify the DAG from a single context?
* Please fix the assumption numbering in the theorem statements.

**Limitations:**

Briefly described in Section 6. No major concerns.

---

> ### Author Rebuttal · Authors · 2023-08-09
>
> Thank you for your time and detailed comments, which we address in the following.
>
> **Identifiability given a single context**
> > The DAG from a single context is identifiable as it is a nonlinear additive noise model.
>
> While the DAG is indeed identifiable for *many* generic functions, the results of Peters (2014) [1] are obtained by assuming differentiability of the underlying distributions as well as causal mechanisms.
> By modeling our causal mechanisms via GPs, these assumptions are not applicable since individual GP paths are in general not differentiable. For our more general setting, we are not aware of proofs that the DAG is identifiable from a single context.
>
> Peters (2014) [1] give examples where the causal direction is not identifiable via independence of residuals (Ex. 2). Given these limitations, it is interesting to explore independence of causal mechanisms as an alternative criterion, and we hope that our work improves the current theoretical understanding.
>
> **Dependency on the number of contexts**
> > Theorem 3.2 seems very unrealistic when considering infinite number of contexts.
>
> Infinite numbers of contexts are of course highly unrealistic, which is why we provide refined bounds in Thm. 3.3. Note also that our results address the case where our data comes from random environments over which we have no control.
> When we do have control over the environments, i.e., when we can actively perform interventions with known effects, then general results on the numbers of required contexts apply, requiring data from $O(\log(m))$ environments for $m$ variables. [2]
>
> > While the authors try to provide some justification to it by providing a finite-sample statement in Theorem 3.3, I got confused with the idea of $C$
>  falling into some "bins".   What does it mean for $C$ to fall into two or more different bins?
>
> We agree that this description is confusing and will improve the explanation. The assumption that the partition sizes are fixed can be understood in a frequentist sense, in that on average each possible causal mechanism for each variable $X_j$ occurs in at least a constant fraction of all samples. For example, if we study the differences between treatments in different hospitals, as we obtain more data, these data would be obtained from the *same* set of hospitals, rather than by the addition of new hospitals.
>
> > The effectiveness of the proposed model heavily depends on certain characteristics of the dataset such as having enough contexts, which might not always be the case.
>
> Note that in Theorem 3.3 we show how many contexts it takes us to identify both the causal graph *as well as* the partition of the causal mechanisms across environments. If we care only about identifiability of the causal graph, then two mechanisms per variable (for those variables which are not already directed by Meek's rules) suffice, which dramatically reduces the number of contexts required. In particular, if we obtain samples with equal probabilities for each mechanism, then we require only 20 contexts to orient up to 10000 edges with probability >0.99, independently of the precise structure of the graph.
> While it is true that these 20 contexts are a lot more than the ~8 required for optimal intervention design for causal learning [1], our results do not require us to have control over the intervention targets, an assumption which would in many cases be *even more unrealistic*.
>
> We also concede that observing a system in multiple contexts at all is an assumption that might not always be realistic. Exactly this point, however, can be an argument for using LINC instead of mechanism shift scores, since LINC already works well given few, or even a single, context, as we observed in our evaluation. As an explanation of this phenomenon, we refer to Marx and Vreeken (2019) [3] who give identifiability results for i.i.d. data with MDL-based scoring functions, via independence of regression residuals.
> In light of this, we do not consider the dependence on the number of contexts a limiting concern in practice.
>
>
> **Other concerns**
> > The proposed solution is relatively complex, particularly concerning the computational cost.
>
> We consider our theoretical insights our main contribution while providing a proof-of-concept implementation. We also see the complexity of the solution as a current limitation, and our efficient heuristics can be further optimized (as you point out, for example, the current solution is slower than MSS in some cases).
>
> > Why is $|E|$ in the lower bound in Theorem 3.3. inside exp? I might be missing something but doesn't this come from the union bound and, hence, shouldn't it be outside exp?
>
> This is not a union bound on probability of the union of some set of events, it is a bound on the intersection of multiple events. In effect, we assume that every edge has to be directed independently of all others. This is in fact a *lower bound* on the probability, since determining the direction of some edges can allow for orienting others using Meek's rules.
>
>
> > [...] the authors could greatly reduce the amount text of somewhat repetitive definitions of the model in Line 69, Assumption 1 and Assumption 2.
>
> Thank you for your suggestions on shortening the problem setting, we will remove Assumption 1 as it is subsumed by Assumption 2.
>
> Please let us know whether we could address your concerns. We are happy to respond to any further questions and concerns.
>
> **References**
>
> [1] Jonas Peters, Joris M. Mooij, Dominik Janzing, and Bernhard Schölkopf. 2014. "Causal discovery with continuous additive noise models." *J. Mach. Learn. Res.* 15, 1 (January 2014), 2009–2053.
>
> [2] Hauser, Alain, and Peter Bühlmann. "Two optimal strategies for active learning of causal models from interventional data." *International Journal of Approximate Reasoning* 55.4 (2014): 926-939.
>
> [3] Marx, Alexander, and Jilles Vreeken. “Identifiability of Cause and Effect Using Regularized Regression.” In *KDD*, 2019.

---

> > ### Comment · Reviewer_t48a · 2023-08-21
> >
> > Thank you for the detailed responses. I did not have further clarification questions. I will make sure to carefully include your responses during the reviewers' discussion period.

---

> > > ### Author Response · Authors · 2023-08-21
> > >
> > > Thank you for your response! We will revise our manuscript according to your comments.

---

### Author Rebuttal · Authors · 2023-08-09

**Summary Response to All Reviewers**

We thank all reviewers for their detailed comments and summarize our response to the main concerns below.

- **Identifiability from a single, few, or many contexts:** We clarify under which conditions the causal DAG is identifiable from a single context to motivate why we study identifiability from multiple contexts (`t48a`). We also state the implications of Thm. 3.3. on the number of contexts needed for DAG identifiability when we have no active control over the interventions (`t48a`), as well as explain the observation that LINC already discovers causal directions given a single context (``LpAE``).

- **Additional explanations**: We provide additional explanations on our assumptions in the individual responses (``LpAE``,``oMbi``, ``NaRE``). We will add these to the manuscript as well as fix our assumption numbering.


We kindly ask the reviewers to take our responses into account in their scores, as we think that the feedback helped us strengthen the presentation of our work. We are happy to respond to follow-up questions.

---

### Decision · Program_Chairs · 2023-09-21

**Decision:**

Accept (poster)

**Comment:**

A majority of reviewers are in favor of accepting this paper, however, the paper contains a critical omission that is needed before this paper can be accepted. In Section 2.2, the concept of algorithmic independence is mentioned, but never defined (in particular, see Postulate 2.1.1). This was explicitly flagged by Reviewer oMbi, while the rest of the reviewers unanimously agreed the assumptions were not clear and the writing was hard to follow. The authors, in their response, promise to "include the definition in the paper", but I fail to see why the definition was not included in the response itself.

In looking this over myself, it is hard to gauge the usefulness of these results without an explicit, formal definition of algorithmic independence. It is worth noting that this concept has multiple definitions in the literature, and so the importance of formalizing this for the present paper cannot be overstated.  Fortunately for the authors, it does not seem that the main theorem statements rely on Postulate 2.1, although I'd like to see this explicitly clarified in the paper. Along these lines and more generally, the writing can be substantially improved, as noted by all the reviewers.

At the same time, a majority of reviewers agree there are some interesting new ideas in this paper, and recommend accept in spite of this omission. On this point I also agree, and am willing to recommend accept conditional on the authors addressing the above issue in their camera ready.